



Climate
of the Past

# Variations in mineralogy of dust in an ice core obtained from northwestern Greenland over the past 100 years

**Naoko Nagatsuka**[1], **Kumiko Goto-Azuma**[1,2], **Akane Tsushima**[3], **Koji Fujita**[4], **Sumito Matoba**[5], **Yukihiko Onuma**[6], **Remi Dallmayr**[7], **Moe Kadota**[5,8], **Motohiro Hirabayashi**[1], **Jun Ogata**[1], **Yoshimi Ogawa-Tsukagawa**[1], **Kyotaro Kitamura**[1], **Masahiro Minowa**[4], **Yuki Komuro**[1], **Hideaki Motoyama**[1,2], **and Teruo Aoki**[1]

[1]National Institute of Polar Research, Tokyo 190-8518, Japan
[2]Department of Polar Science, The Graduate University for Advanced Studies, SOKENDAI, Tokyo 190-8518, Japan
[3]Graduate School of Science, Chiba University, Chiba 277-0882, Japan
[4]Graduate School of Environmental Studies, Nagoya University, Nagoya 464-8601, Japan
[5]Institute of Low Temperature Science, Hokkaido University, Sapporo 060-0819, Japan
[6]Institute of Industrial Science, University of Tokyo, Kashiwa 277-8574, Japan
[7]Alfred Wegener Institute, Helmholtz-Zentrum für Polar- und Meeresforschung, Bremerhaven, Germany
[8]Graduate School of Environmental Science, Hokkaido University, Sapporo 060-0810, Japan

**Correspondence:** Naoko Nagatsuka (nagatsuka.naoko@nipr.ac.jp)

**Abstract.** Our study is the first to demonstrate a high-temporal-resolution record of mineral composition in a Greenland ice core over the past 100 years. To reconstruct past variations in the sources and transportation processes of mineral dust in northwestern Greenland, we analysed the morphology and mineralogical composition of dust in the SIGMA-D ice core from 1915 to 2013 using scanning electron microscopy (SEM) and energy-dispersive X-ray spectroscopy (EDS). The results revealed that the ice core dust consisted mainly of silicate minerals and that the composition varied substantially on multi-decadal and inter-decadal scales, suggesting that the ice core minerals originated from different geological sources in different periods during the past 100 years. The multi-decadal variation trend differed among mineral types. Kaolinite, which generally formed in warm and humid climatic zones, was abundant in colder periods (1950–2000), whereas mica, chlorite, feldspars, mafic minerals, and quartz, which formed in arid, high-latitude, and local areas, were abundant in warmer periods (1915–1949 and 2005–2013). Comparison to Greenland surface temperature records indicates that multi-decadal variation in the relative abundance of these minerals was likely affected by local temperature changes in Greenland. Trajectory analysis shows that the minerals were transported mainly from the western coast of Greenland in the two warming periods, which was likely due to an increase in dust sourced from local ice-free areas as a result of shorter snow/ice cover duration in the Greenland coastal region during the melt season caused by recent warming. Meanwhile, ancient deposits in northern Canada, which were formed in past warmer climates, seem to be the best candidate during the colder period (1950–2004). Our results suggest that SEM–EDS analysis can detect variations in ice core dust sources during recent periods of low dust concentration.

## 1 Introduction

Aeolian mineral dust in snow and ice on ice sheets provides key information about global and local climate change. Past ice core dust records have revealed substantial variations in the concentration, composition, particle size, and morphology of minerals on glacial–interglacial timescales (ca. 800 kyr BP; e.g. Petit et al., 1990; Lambert et al., 2008), geologic timescales (from the Eemian to the Holocene; e.g. Maggi, 1997; Ram and Koenig, 1997; Steffensen, 1997; Ruth et al., 2003; Schüpbach et al., 2018; Simonsen et al., 2019), and seasonal scales (e.g. Bory et al., 2003a; Drab et al., 2002). Ice core dust records show a close relation-

ship with temperature changes and atmospheric circulation. Dust concentrations in Greenland ice cores during the last glacial period were 10 to 100 times higher than those over the Holocene and were strongly correlated with temperature changes (as indicated by $\delta^{18}O$ records; e.g. De Angelis et al., 1997; Mayewski et al., 1997; Fuhrer et al., 1999; Ruth et al., 2003 TS2; Schüpbach et al., 2018). Steffensen TS3 (1997) also showed a systematic connection between dust volume distribution, total dust mass, and $\delta^{18}O$ in the Greenland Ice Core Project (GRIP) ice core, indicating that climate changes appear to have modified the processes of formation, transport, and deposition of mineral dust in the same way over the last 120 000 years. Meanwhile, there is a significant correlation between the dust flux of the European Project for Ice Coring in Antarctica (EPICA) ice core with Antarctic temperature during glacial periods, but not during interglacial periods, suggesting that the conditions of source areas and the transport of ice core dust change according to the glacial–interglacial cycle (Lambert et al., 2008). The variability of ice core dust may be related to changes in atmospheric transport and dust source areas affected by climate change (Svensson et al., 2000). Thus, it is important to reconstruct the variations in the sources and transportation processes of mineral dust in ice cores.

Geochemical analyses, such as stable isotope ratios of Sr, Nd, and Pb, have been used to identify possible sources of Greenland ice core dust. These isotopic ratios have strong regional variations that are controlled by their geological origins and are hardly altered during transportation in the atmosphere or after deposition (Capo et al., 1998; Faure and Mensing, 2004). The isotope ratios of the GRIP and Greenland Ice Sheet Project 2 (GISP2) ice core dust obtained from 44 to 12 kyr BP indicated that eastern Asian deserts appeared to be the most likely dust sources (Biscaye et al., 1997; Svensson et al., 2000) and that central/east central European loess might also be a major source of dust in the last glacial ice core (Újvári et al., 2015). Lupker et al. (2010) analysed the Sr and Nd isotopic ratios of mineral dust in an ice core from southern Greenland (Dye 3) in the age range of 1786–1793 CE and revealed that the Sahara might be an additional dust source. Han et al. (2018) showed temporal variations in the source areas of the NEEM ice core dust based on Sr and Pb isotope analysis; the primary dust source was eastern Asian deserts from 31 to 23 kyr and the Sahara from 23 to 12 kyr. However, these analyses have targeted mostly ice core dust from glacial periods characterized by a high dust concentration because Sr and Nd isotopic ratio analyses need large numbers of samples. Although some studies have analysed the Sr and Nd isotopes of ice core minerals during the Holocene, a recent period of low dust concentration, they needed to concentrate decades to thousands of years of ice for each sample (e.g. Bory et al., 2003a; Han et al., 2018; Simonsen et al., 2019). Thus, there is limited information about possible sources of mineral dust in interglacial periods during which dust concentrations are low.

Scanning electron microscopy (SEM) and energy-dispersive X-ray spectroscopy (EDS) are useful tools for revealing the source areas of ice core dust samples with low amounts of mineral dust. SEM provides morphological information, and EDS provides the mineralogical composition of individual particles, allowing the evaluation of the continental dust input and showing variations in ice core dust properties. Donarummo et al. (2003) analysed the size distribution and mineral composition of GISP2 ice core dust during the 1930s by SEM–EDS and suggested that the central United States might have contributed a substantial amount of minerals to the ice core when the source area was affected by intense droughts. The SEM–EDS analysis of dust from snow pit samples from 1989 to 1991 at Summit in central Greenland indicated that the possible sources were likely to be Asian deserts, and the source areas have not changed seasonally (Drab et al., 2002). Therefore, SEM–EDS analysis can demonstrate a high-temporal-resolution record of composition and sources of ice core minerals during interglacial periods. However, continuous variations in the dust properties of Greenland ice cores during recent years are still not well known.

Possible source areas for the Greenland ice core dust may have varied in recent years. Most of the Earth's surface has changed rapidly because of recent climate warming and human activities, which may result in changes in atmospheric transport and sources of mineral dust to the ice sheet. For example, dust outbreaks occurring in eastern Asian deserts, which are vast sources of aeolian mineral dust, remarkably increased from 2000 to 2002 compared with the 1990s (Kurosaki and Mikami, 2003). The dust from Asian deserts is transported to Greenland across the ocean. Zhang et al. (2020) revealed that there has been an abrupt shift toward a hotter and drier climate over east Asia over the past 260 years, according to tree-ring-based reconstructions of heat waves and soil moisture. Furthermore, there may also be an increasing contribution of dust from local source areas in Greenland (Amino et al., 2020). The retreating ice and decreasing seasonal snow will expose more sediment in the proglacial area, delivering greater quantities of fine sediments to the floodplain than at present. Bullard and Austin (2011) reported that exposure of the proglacial floodplain in Kangerlussuaq, western Greenland, during ice retreat may also make more material available for aeolian transport.

This paper aims to describe the temporal variations in sources of minerals in a Greenland ice core covering a nearly 100-year period (1915–2013) with a 5-year resolution, during which the Arctic region was remarkably warming. The morphology and mineralogical composition of the ice core dust were analysed by SEM and EDS, and variations are discussed in terms of changes in the ice core dust sources.

Please note the remarks at the end of the manuscript.

Clim. Past, 17, 1–22, 2021                                                                    https://doi.org/10.5194/cp-17-1-2021

## 2 Samples and analytical methods

### 2.1 SIGMA-D ice core

The ice core was drilled at 2100 m a.s.l. in an accumulation area of the northwestern Greenland Ice Sheet called the SIGMA-D site (77.636° N, 59.120° W) in May 2014 (Matoba et al., 2015). The SIGMA-D site is located 250 km east of the town of Qaanaaq and lies in the upstream section of Heilprin Glacier, the largest outlet glacier in this area (Fig. 1). The ice core was recovered from the surface down to a depth of 222.72 m (total core length).

### 2.2 Water isotope and ion concentration measurement

To determine the annual layers of the SIGMA-D ice core, the stable isotopes of water ($\delta^{18}O$) and the concentrations of sodium and sulfate ions ($Na^+$ and $SO_4^{2-}$) were measured at the Institute of Low Temperature Science, Hokkaido University, and the concentration of tritium was measured at the National Institute of Polar Research (NIPR), Tokyo, Japan.

The ice core samples were cut into 5–10 cm pieces with cross sections of 30 cm$^2$ and decontaminated by removing the surface of each sample with a ceramic knife. Then, each sample was placed in a clean polyethylene bag and melted in a water bath, before being transferred to pre-cleaned polypropylene bottles in a tent for sample preparation at the field camp (Matoba et al., 2015). $\delta^{18}O$ was measured using a near-infrared cavity ring-down spectrometer (IR-CRDS, Picarro L2130-i, USA) with a high-throughput Picarro-A0212 vaporizer. The precision of determination was ±0.08 ‰ for $\delta^{18}O$. The concentrations of $Na^+$ and $SO_4^{2-}$ were determined by ion chromatography (ICS-2100, Thermo Fisher Scientific, USA). Dionex AS-14A and CS-12A columns (Thermo Fisher Scientific) were used for anion and cation analyses, respectively. The limit of quantification was 5 ppb for both ions. For $SO_4^{2-}$, we also calculated its non-sea-salt (nss) fractions as follows:

$$[nssSO_4^{2-}] = [SO_4^{2-}] - (SO_4^{2-}/Na^+)_{sea} \times [Na^+], \tag{1}$$

where $(SO_4^{2-}/Na^+)_{sea}$ is the mass ratio of $SO_4^{2-}$ to $Na^+$ in the seawater, which is 0.252 (Wilson, 1975; Legrand and Mayewski, 1997). Tritium concentrations were measured using a liquid scintillation counter (LSC-LB3; Aloka Co. Ltd., Japan). The vertical resolution of the tritium measurements was 0.5 m.

### 2.3 Dust concentration

The concentration of dust in the SIGMA-D ice core was measured using an Abakus laser particle sensor (Klotz GmbH, Germany) connected to a continuous-flow analysis (CFA) system, a slightly modified version of the one reported by Dallmayr et al. (2016). The size bins cover the range from 1.5 to 15.0 µm, and the depth resolution of the measurement was 2–3 cm. CFA analysis has not been performed since 2003 (above a depth of 6.35 m) due to poor core quality.

### 2.4 SEM–EDS analysis of mineral dust

To extract mineral particles from the ice core at 5-year intervals, 4 cm$^2$ cross sections were cut from the 50 cm long archived core sections in the −20 °C cold room at NIPR. The possibly contaminated outer layers (∼ 1 cm thick) were removed using a pre-cleaned ceramic knife. Then, several millimetres of the ice surface was scraped off by the ceramic knife and collected in a clean 100 mL polyethylene bottle for every 5-year interval. The samples were freeze-dried at −45 °C using a freeze dryer (DRW240DA; Advantec, Japan) on a polycarbonate membrane filter (Advantec) with a diameter of 25 mm and pore size of 0.1 µm. The morphological characteristics and chemical composition of individual mineral particles on the membrane filter were observed by SEM (Quanta FEG 450, FEI) combined with EDS (X-Max 50, Oxford Instruments, UK) at NIPR. The filter targets were mounted on aluminium stubs using double-faced adhesive carbon tape and coated with vaporized platinum for SEM observation. In total, 150 particles were randomly chosen from the filter, and the equivalent circle diameter, the two-dimensional area ($A$), and the perimeter ($P$) were measured on digital photographs with an image-processing application (ImageJ, National Institutes of Health, USA). Then, the shape parameters of the particles were obtained, namely circularity ($= (4\pi A) \times P^{-2}$). The major elemental composition (Na, Mg, Al, Si, Cl, S, Ca, K, Fe, P, and Ti) and related oxides ($Na_2O$, $MgO$, $Al_2O_3$, $SiO_2$, $CaO$, $K_2O$, $Fe_2O_3$, $P_2O_5$, and $TiO_2$) were obtained from the EDS spectra. The acceleration voltage and working distance for SEM analysis were 20 kV and 10 mm, respectively, and each EDS spectrum had more than 100 000 acquisition counts. To be counted as a mineral dust, a particle had to contain at least one of the elements Na, Mg, Si, Al, K, Ca, and Fe, each with an atomic ratio (%) amount at least twice that of the error (%). We did not count soluble particles, such as $CaSO_4$, $Na_2SO_4$, and $NaCl$, that can be derived from volcanic and marine aerosols.

### 2.5 Mineral identification

The mineralogical identification was performed using elemental composition and related oxides of individual mineral particles. Previous studies used protocols to semiquantitatively identify the mineralogy of individual particles in ice cores by SEM–EDS analysis (e.g. Mudroch et al., 1977; Maggi, 1997; Donarummo et al., 2003; Wu et al., 2016). The identification of the SIGMA-D ice core dust followed three procedures (Wu et al., 2016). First, the spectrum pattern of each particle was matched to those of standard minerals (Severin, 2004). Second, we compared the oxide composition and morphology of the ice core dust with those of the standard minerals. Finally, a sorting scheme used

https://doi.org/10.5194/cp-17-1-2021

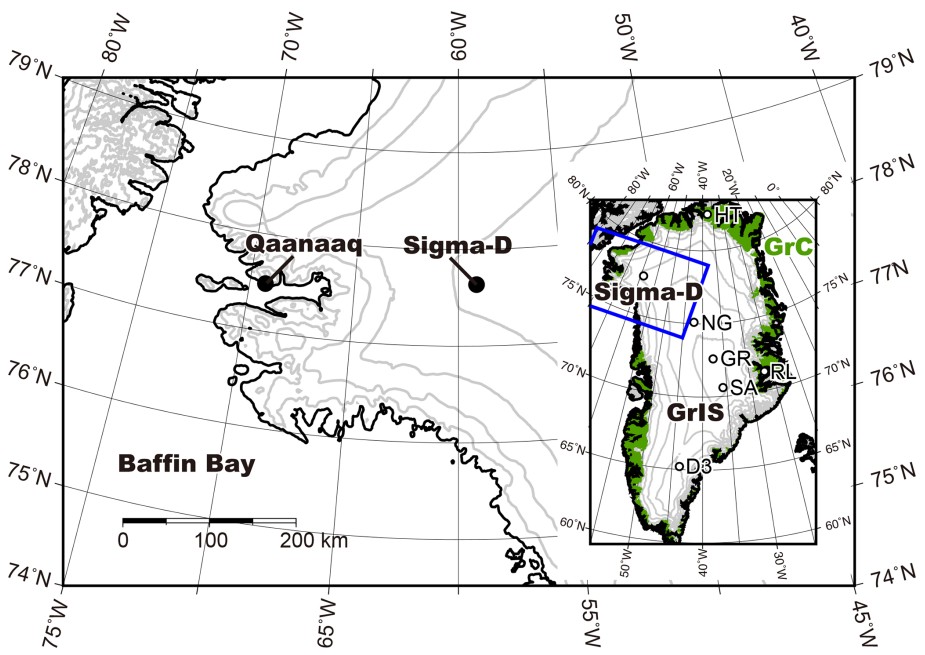

**Figure 1.** Location map of the SIGMA-D site (77.636° N, 59.120° W, 2100 m a.s.l.). Contour lines are drawn at a 500 m interval. The blue frame in the inset map of Greenland (GrIS) denotes the domain of the main map. The green shaded region in the inset map denotes the ice-free coastal terrain (GrC), and HT, NG, GR, RL, SA, and D3 denote Hans Tausen, NGRIP, GRIP, Renland, Site-A, and Dye-3 ice core sites for the back-trajectory comparison, respectively.

to identify minerals in the GISP2 Greenland ice core dust by SEM–EDS was applied to identify the mineral types from the peak intensity ratios (Donarummo et al., 2003). Comparing the results of these procedures enables reliable mineral identification (Maggi, TS5 1997; Wu et al., 2016).

Based on the formation process (weathering types), formation environment (temperature and humidity), and possible sources of the SIGMA-D ice core dust, most of the silicates we analysed could be classified into the following five types. Type A consists primarily of kaolinite, which is a clay mineral generally formed by chemical weathering in warm and humid regions, including Africa, South America, and Southeast Asia (e.g. Mueller and Bocquier, 1986; Velde, 1995; Bergaya et al., 2006). We also found a mineral composed of Si and Al, but with a higher proportion of Si and a lower proportion of Al compared with kaolinite. It is likely pyrophyllite, which is generally found with kaolinite, and is thus also classified as Type A. Type B comprises mica, chlorite, and a mixture of the two, which are clay minerals formed by mechanical weathering of igneous and metamorphic rocks in cold and dry regions (e.g. Cremaschi, 1987; Pye, 1987; Velde, 1995). Type C consists of feldspars (Na/Ca-plagioclase and K-feldspar), which are also formed by mechanical weathering in cold and dry regions (e.g. Nahon, 1991). Type D consists of mafic minerals containing abundant Mg and Fe, such as hornblende and pyroxene, which are less frequent in atmospheric dust and are formed by mechanical weathering (e.g. Deer et al., 1993). Type E

consists of quartz, which are the most physically and chemically resistant minerals to weathering, and their abundance in the atmosphere is related to the large desert source areas (Pye, 1987; Yokoo et al., 2004 TS6; Genthon and Armengaud, 1995). According to previous studies, some minerals have localized distributions. Ito and Wagai (2017) showed the global terrestrial distribution of clay size mineral groups, revealing that Type A minerals were predominant in humid regions in low- or middle-latitude areas, whereas Type B minerals were abundant in arid and/or high-latitude areas. Furthermore, kaolinite (Type A) can be used as an indicator of intensive weathering in palaeoclimatic conditions (Biscaye, TS7 1965; Griffin et al., 1968). For example, the relative abundance of kaolinite (Type A) to chlorite (Type B) is a mineral indicator that is most sensitive to latitude dependency. The kaolinite / chlorite ratio shows higher values for minerals from low latitudes, such as North Africa, but showed lower values for minerals from the Northern Hemisphere, such as Asia and North America (e.g. Biscaye et al., 1997; Maggi, TS8 1997; Svensson et al., 2000; Donarummo et al., 2003). This trend reflects a decrease in weathering intensity with latitude. In addition to variations in kaolinite and chlorite, Type C, D, and E minerals also reflect the geological and climatic conditions of their source areas, as mentioned above. Thus, compositional variations among the five types of minerals can be used as an indicator of the source and transportation process of ice core dust in different periods. In this study, we consider each type of mineral as possibly being contributed from the

https://doi.org/10.5194/cp-17-1-2021

following sources: Type A, low- to mid-latitude areas (e.g. central Africa, South America, Southeast Asia); Types B and C, high-latitude (e.g. North America, Russia, north Europe) and/or desert areas (e.g. Asia and North Africa); Type D, local areas (Greenland); and Type E, desert areas (e.g. Asia and North Africa).

## 2.6 Backward trajectory analysis

To investigate the possible source regions of the ice core dust, the air mass transport pathways were analysed using the Hybrid Single-Particle Lagrangian Integrated Trajectory (HYSPLIT) model, which is distributed by the National Oceanographic and Atmospheric Administration (NOAA; Stein et al., 2015). Points at 50, 500, 1000, and 1500 m above ground level (a.g.l.) at the SIGMA-D site were set as the initial points of an air mass for the 7 d backward trajectories. The probability distribution of the air mass at altitudes below 1500 m a.g.l. was calculated at a 1° resolution. We assumed wet and dry deposition processes for the preserved aerosol tracers (Iizuka et al., 2018; Parvin et al., 2019). For wet deposition, the probability was weighted by the daily precipitation when the air mass arrived at the ice core site. For dry deposition, the probability was based on the counting of non-precipitation days when the air mass arrived. We used daily precipitation from the ERA-40 and ERA-Interim reanalysis datasets, both of which were produced by the European Centre for Medium-Range Weather Forecasts (Dee et al., 2011; Uppala et al., 2005). The daily precipitation of ERA-40 ($p_{40}$) was calibrated with that of ERA-Interim ($p_{int}$) via linear regression obtained for the period 1979–2001 ($p_{int} = 0.47 p_{40}$, $R^2 = 0.702$, $p < 0.001$) to maintain consistency between the two precipitation datasets for the entire period (1958–2013). We also calculated the regional contribution from the probability distribution, for which land regions were divided into the following five regions: the Greenland Ice Sheet, the Greenland coast, Canada (including Alaska), northern Eurasia, and mid-latitude arid regions (consisting of China, Central Asia, and the Middle East) (Fig. 2a).

To compare the sources of mineral dust in the Greenland Ice Sheet, the air mass transport pathways were analysed not only from the SIGMA-D site, but also from the other six Greenland ice core sites for which Bory et al. (2003b) revealed the dust sources based on Sr and Nd isotope ratios; the four interior sites (NGRIP, GRIP, Site-A, and Dye-3) contain more dust from eastern Asian deserts compared with the two coastal sites (Hans Tausen and Renland). We analysed the 7 d back trajectories of the air masses for 1981–2010 with an initial height of 500 and 1500 m a.g.l. at the ice core sites.

## 2.7 Snow cover fraction

To examine the surface conditions of neighbouring source areas of mineral dust, we analysed inter-annual changes in snow cover fraction derived from multiple numerical simulations by a climate model. Various international organizations have used global climate models to conduct numerical simulations that reproduce or predict climate change from past to future. The results have been published by Coupled Model Intercomparison Project Phase 6 (CMIP6; Eyring et al., 2016) under the auspices of the World Climate Research Programme. In the present study, snow cover fractions derived from historical simulations for the period 1850–2014 (Onuma and Kim, 2020a, b, c, d) were used to examine mineral dust sources for SIGMA-D. The dataset was produced by MIROC6, a climate model developed by a Japanese modelling community (Tatebe et al., 2019). Four reanalysis datasets, namely GSWP3 (Kim, 2017), CRUJRA (Harris, 2019), Princeton (Sheffield et al., 2006), and WFDEI (Weedon et al., 2014) provided the meteorological conditions for the Land Surface, Snow and Soil moisture Model Intercomparison Project (LS3MIP; van den Hurk et al., 2016), which is a sub-project of CMIP6. We obtained data on inter-annual changes in snow cover fraction during summer on the northwest and southwest coasts of Greenland (boundary at 70° N).

## 3 Results

### 3.1 Dating of the SIGMA-D ice core

Dating of the SIGMA-D ice core was performed by annual layer counting of $\delta^{18}$O and Na$^+$ that showed obvious seasonal variations (Fig. A1). The observed seasonality of chemical components and the water stable isotope ratio in the snowpack and ice cores have previously been reported at various sites on the Greenland Ice Sheet (Whitlow et al., 1992; Legrand TS10 and Mayewski, 1997; Kuramoto et al., 2011; Oyabu et al., 2016; Kurosaki et al., 2020). The winter season of the SIGMA-D ice core was defined as the depth at which $\delta^{18}$O was at its minimum value and Na$^+$ was at its maximum value, and we counted winter season to winter season as 1 year.

Other fixed dates were provided by the tritium profile and nssSO$_4^{2-}$ spikes. A sharp tritium peak at 11.56 m w.e. corresponds to the H-bomb test in 1963 (Koide et al., 1982, Clausen and Hammer, 1988), indicating an accumulation rate of 0.23 m w.e. yr$^{-1}$ from 1963 to 2013. The large nssSO$_4^{2-}$ peak appearing at 54.17 m w.e. is assumed to correspond to the eruption of the Laki volcano in 1783. The nssSO$_4^{2-}$ signal of the 1783 Laki eruption has also been found in other ice cores in Greenland, Arctic Canada, and Svalbard (Clausen and Hammer, 1988; Grumet et al., 1998; Matoba et al., 2002). Similarly, we assume other nssSO$_4^{2-}$ spikes to be the signatures of unknown (1810), Tambora (1816), and Katmai (1912) volcanic eruptions at shallower depths of 47.53, 46.03, and 23.50 m w.e., respectively. Comparing the annual layer counting and these reference horizons, we estimate that the ice core dating includes a 1-year error.

As a result of these analyses, we estimate that the upper 112.87 m (86.06 m w.e.) of the ice core is equivalent to the

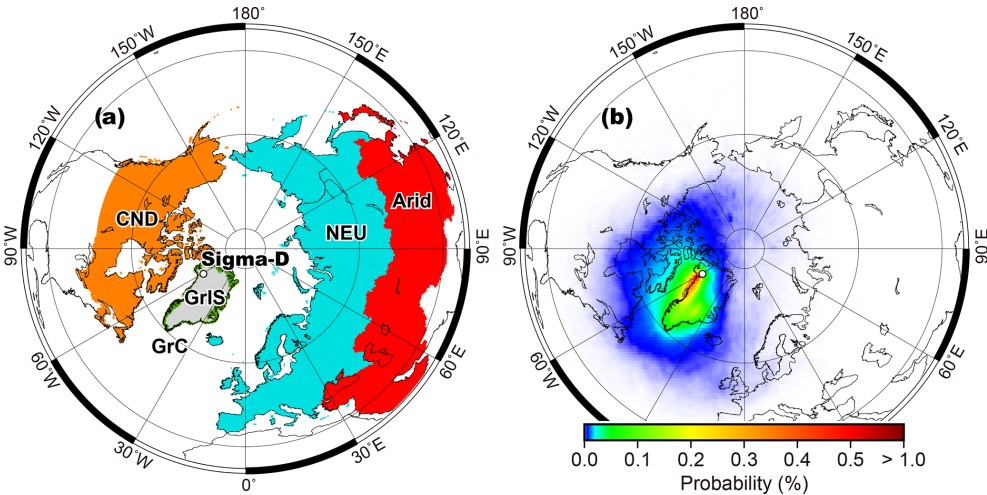

**Figure 2.** TS9 Map showing **(a)** the location of the SIGMA-D (Sigma-D) ice core site in Greenland and five regions for calculating regional contribution (GrIS: Greenland Ice Sheet, grey; GrC: Greenland coast, green; CND: Canada and Alaska, orange; NEU: northern Eurasia, light blue; Arid: arid regions including China, Central Asia, and the Middle East, red) and **(b)** the probability distribution of an air mass at the SIGMA-D site from a 7 d three-dimensional back-trajectory analysis from 1958 to 2014.

period from 1660 to 2013. In this study, we used ice samples to a depth of 38.60 m (22.75 m w.e.) covering 1915 to 2013 for the SEM and EDS analyses (Table 1).

## 3.2 Particle morphology

Figure 3 shows SEM images of the mineral dust in the SIGMA-D ice core. The number size distribution of mineral dust in the SIGMA-D ice core showed that most particles had a diameter of < 2 µm (Fig. 4), which is consistent with other Greenland ice core dust (e.g. Steffensen, 1997; Biscaye et al., 1997). The mean and maximum particle diameters, calculated as 5-year averaged values, ranged from 0.97 to 2.60 and 4.94 to 26.51 µm, respectively, with a single modal structure at the peak ranging from 0.35 to 1.15 µm. The size distribution varied among the samples collected from different periods, showing a narrower peak with finer mode (0.35–0.53 µm) for the samples from 1965 to 1979 and a broader peak with a coarser mode for the samples from 1920 to 1924 and 1945 to 1949 (0.97–1.15 µm; Fig. A2). There were coarser particles with diameters of > 10 µm in the samples from 1915 to 1959 and from 1990 to 2013, but no particles with diameters of > 10 µm were found in the samples from 1960 to 1989, except for the sample from 1980 to 1984.

## 3.3 Quantitative estimation of mineral dust

The elemental composition of individual mineral particles obtained from the EDS analysis showed that the ice core dust was composed mainly of silicate minerals in all the samples (65 %–95 %, Fig. 6). Based on a peak intensity ratio sorting scheme (Donarummo et al., 2003) and comparison of oxide composition and morphological information with those of standard minerals, the silicates were categorized as quartz, Na/Ca- and K-feldspars, clays (kaolinite, pyrophyllite, smectite, illite, mica, and chlorite, as well as mixed layers of illite–smectite and mica–chlorite), and mafic minerals rich in magnesium and iron (Figs. 3 and 7). These minerals were also found in Greenland ice cores from glacial periods (e.g. Maggi, 1997; Svensson et al., 2000).

Semi-quantitative analysis of the EDS spectrum showed that the proportion of kaolinite among 150 mineral particles found in each sample was the highest (5 %–66 %) and that of smectite was the lowest (0 %–2 %) in nearly every period (Fig. 7 and Table 2). The proportion of pyrophyllite, mica–chlorite mix, quartz, feldspars, and illite–smectite mix was the second highest, varying from 6 %–25 %, 3 %–25 %, 3 %–22 %, 1 %–20 %, and 0 %–13 %, respectively. The mineralogy of the SIGMA-D ice core dust showed significantly higher kaolinite contents compared with those of the other Greenland ice cores (GRIP, 4 %–16 %, Svensson et al., 2000; GISP2, 0 %–2 %, Donarummo et al., 2003).

The silicate mineral composition showed large variations among the samples on two different timescales. First, the compositions varied on a multi-decadal scale, with higher kaolinite and pyrophyllite contents (30 %–67 % and 7 %–26 %) and lower mica–chlorite mix contents in the 1950 to 2004 samples (3 %–9 %, 1 %–13 %, and 5 %–15 %, respectively), especially in the 1970 to 2004 samples. The opposite trend was observed in the 1915 to 1949 and 2005 to 2013 samples (kaolinite and pyrophyllite: 5 %–20 % and 6 %–16 %, respectively; mica–chlorite mix: 10 %–25 %). The compositional variation also showed higher feldspars, mafic, and quartz contents in the 1915 to 1945 and 1990 to 2013 samples (feldspars: 1 %–20 %; mafic: 0 %–9 %; quartz: 10 %–22 %) than in the other periods (feldspars: 3 %–6 %;

**Table 1.** Description of the SIGMA-D ice core dust samples.

| Period | Ice | | Dust (size) | | |
|---|---|---|---|---|---|
| | Top (m) | Bottom (m) | Average (µm) | Maximum (µm) | Log-normal mode (µm) |
| 1915–1919 | 37.00 | 38.60 | 2.00 | 12.80 | 0.73 |
| 1920–1924 | 35.49 | 37.00 | 2.60 | 16.30 | 1.15 |
| 1925–1929 | 33.80 | 35.49 | 2.20 | 26.50 | 0.46 |
| 1930–1934 | 31.85 | 33.80 | 1.60 | 12.10 | 0.58 |
| 1935–1939 | 30.22 | 31.85 | 2.30 | 20.10 | 0.59 |
| 1940–1944 | 28.57 | 30.22 | 1.70 | 15.00 | 0.59 |
| 1945–1949 | 26.87 | 28.57 | 2.60 | 21.80 | 0.97 |
| 1950–1954 | 25.02 | 26.87 | 1.50 | 12.80 | 0.55 |
| 1955–1959 | 23.67 | 25.02 | 2.02 | 14.10 | 0.75 |
| 1960–1964 | 21.89 | 23.67 | 1.24 | 6.70 | 0.60 |
| 1965–1969 | 19.95 | 21.89 | 1.31 | 9.60 | 0.53 |
| 1970–1974 | 17.85 | 19.95 | 0.97 | 4.94 | 0.35 |
| 1975–1979 | 16.30 | 17.85 | 1.02 | 5.50 | 0.43 |
| 1980–1984 | 14.61 | 16.30 | 1.90 | 11.90 | 0.58 |
| 1985–1989 | 12.50 | 14.61 | 1.20 | 5.50 | 0.47 |
| 1990–1994 | 10.03 | 12.50 | 2.20 | 15.90 | 0.68 |
| 1995–1999 | 7.56 | 10.03 | 2.08 | 14.86 | 0.77 |
| 2000–2004 | 4.62 | 7.56 | 1.45 | 8.70 | 0.65 |
| 2005–2009 | 2.31 | 4.62 | 1.86 | 11.29 | 0.60 |
| 2010–2013 | 0.00 | 2.31 | 2.03 | 25.95 | 0.51 |

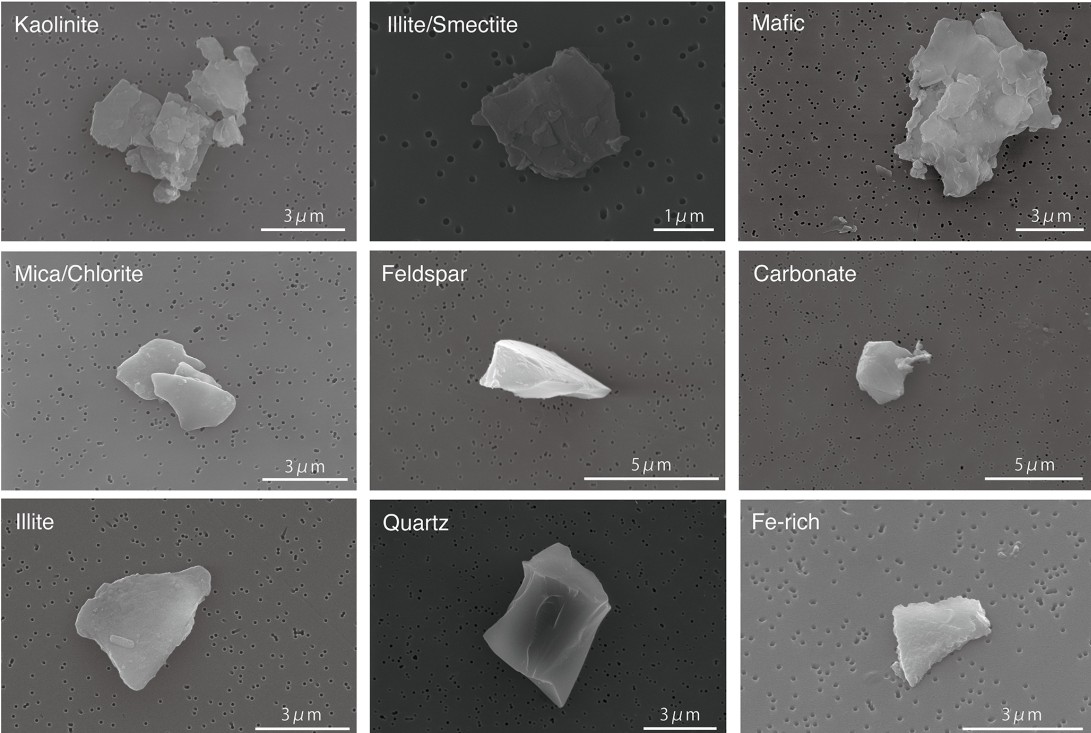

**Figure 3.** SEM images of each mineral group in the SIGMA-D ice core.

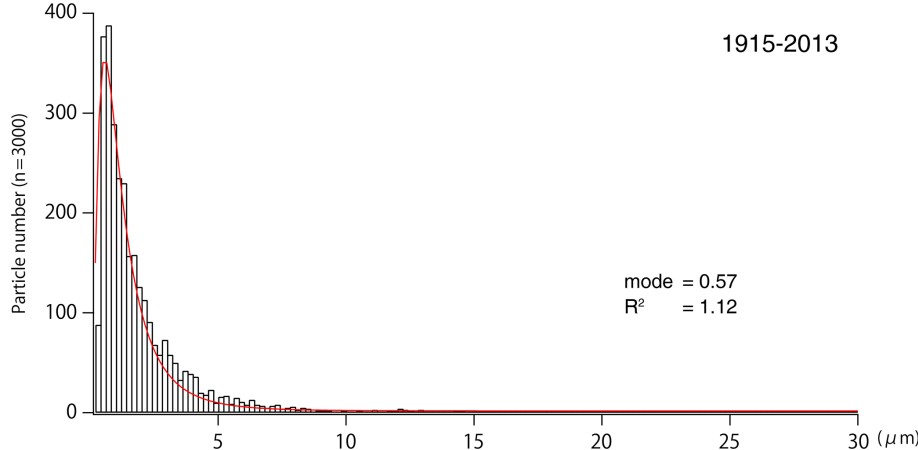

**Figure 4.** Particle size distribution and log-normal fitting results (mode: mode diameter and R2: half peak width) of minerals in the ice core samples from 1915 to 2013.

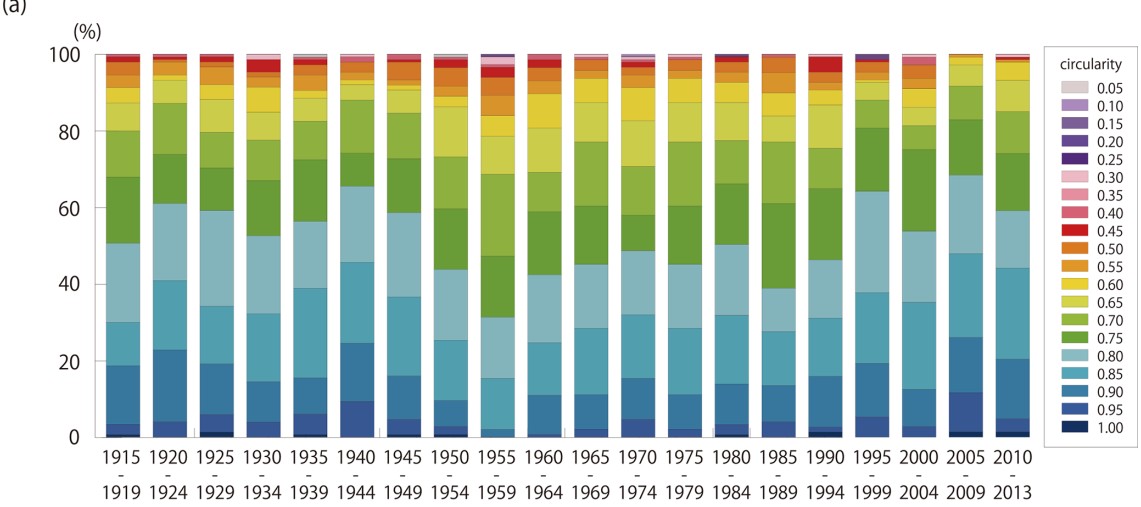

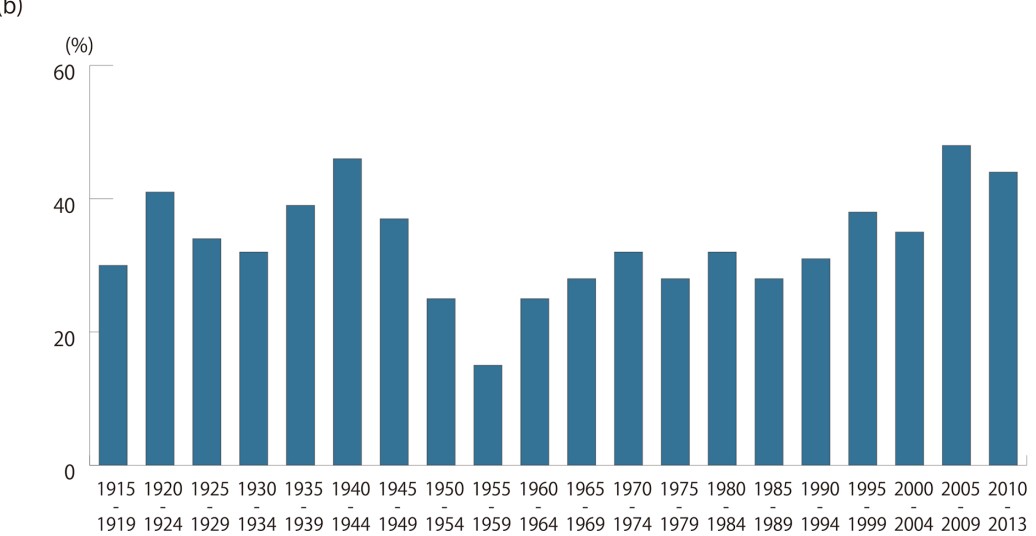

**Figure 5. (a)** Circularity distribution of mineral particles from different period. **(b)** Variation in proportion of circularity values of > 0.8.

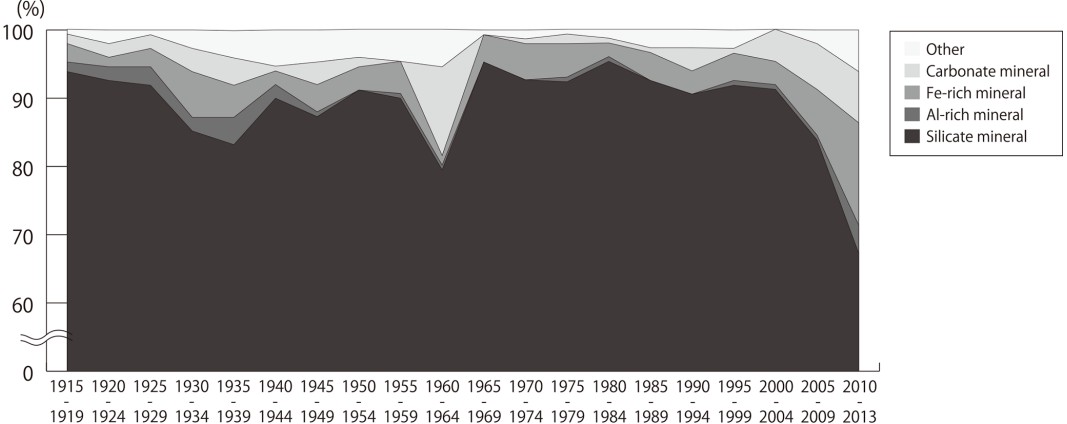

**Figure 6.** Variations in the insoluble mineral records in the ice core dust in 5-year resolution.

mafic: 1 %–2 %; quartz: 3 %–6 %). Second, the compositions varied on shorter timescales, in inter-decadal cycles. Variations in the compositions of feldspars, mica, chlorite, and mica–chlorite mix were similar, but opposite to those of kaolinite and pyrophyllite.

Non-silicate minerals were also found in the ice core samples and were composed mainly of Ca- and Fe-dominant minerals, identified as carbonates (calcite) and Fe-oxides (pyrite, magnetite, or hematite), respectively (Fig. 6). The relative abundance of both minerals, ranging from 0 % to 15 %, has increased in the last 10 years. The carbonate minerals showed the highest abundance in the 1960–1964 sample.

### 3.4 Source regions of SIGMA-D ice core dust

To identify the source regions of the ice core dust, we applied the HYSPLIT back-trajectory model and calculated the probability distributions of an air mass arriving at the SIGMA-D site from 1958 to 2014 (Figs. 2b and 8). The results show that the air mass at elevations above ground level from 0 to 1500 m came mainly from the western coast of Greenland, including the Baffin Bay, whereas a smaller part came from northern Canada (Figs. 2b and 8a). Excluding the ice sheet and ocean areas that could not be possible sources of mineral dust, the air mass is considered to have come mainly from the Greenland coast (50 %–60 %) and Canada ($\sim$ 40 %), with a small contribution from northern Eurasia ($\sim$ 3 %, Fig. 8b). Contributions from these three possible source regions show little seasonal and inter-annual variabilities (Fig. 8b and c). The air mass contribution from the Greenland coast was larger in dry deposition than wet deposition during summer, which may have caused an increase in dust sourced from local ice-free areas. However, there was no significant difference in the overall trend between the two deposition processes.

## 4 Discussion

### 4.1 Variation in silicate mineral composition

The SEM–EDS analysis revealed that the SIGMA-D ice core dust samples collected from 1915 to 2013 contained mainly silicate minerals, which is the most abundant family of crustal minerals (Deer et al., 1993). Silicate mineral composition showed variations on a multi- and inter-decadal scale, indicating that the ice core minerals originated from different geological sources in different periods during the past 100 years.

Variation trends in the silicate mineral composition of the SIGMA-D ice core samples substantially differed among mineral types (Fig. 9), indicating that the minerals in the ice core were derived from multiple geological sources. The dominance of Type A minerals in the samples from 1950 to 2004 indicated that the minerals might be derived mainly from low- or middle-latitude areas in the periods. In contrast, the abundance of Type B, C, D, and E minerals in the samples from 1915 to 1949 and from 2005 to 2013 indicated that the minerals were likely derived from arid deserts and/or high-latitude areas, including Greenland.

The morphological characteristics of the ice core dust also support the changes in the sources of silicate minerals. The size distribution of the minerals showed lower mean, maximum, and modal diameters from 1950 to 1994 compared with the other periods, when the samples consisted mainly of Type A minerals (Table 1, Fig. A2). The circularity also showed a similar trend, containing smaller amounts of particles with circularity values > 0.80 from 1950 to 1994 (Fig. 5). The particle size depended on the mineralogy of the ice core dust. The coarser fraction (> 2 μm) of the ice core dust samples contained little clay, especially for Type A minerals (Fig. 10), but contained an abundance of Type C, D, and E minerals. This particle size dependence is consistent with other analyses of Greenland ice core dust (Biscaye, 1965; Svensson et al., 2000). Thus, the ice core dust was likely

https://doi.org/10.5194/cp-17-1-2021 Clim. Past, 17, 1–22, 2021

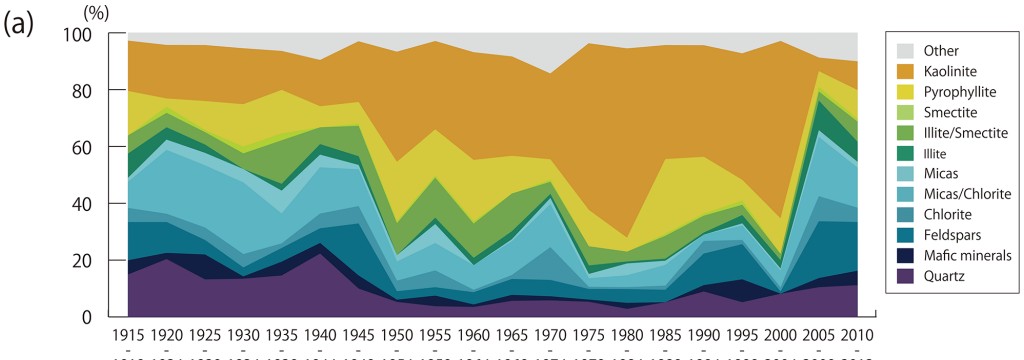

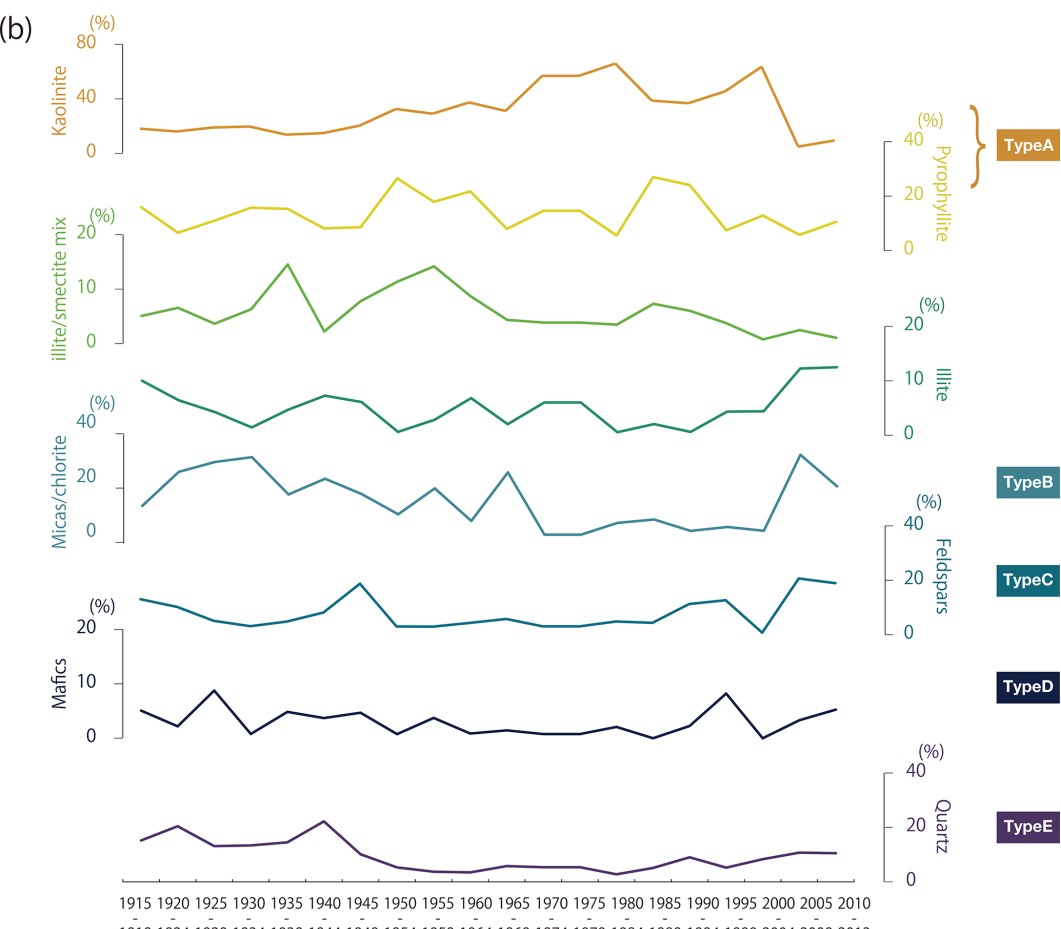

**Figure 7.** Variations in the silicate mineral records in the ice core dust in 5-year resolution. **(a)** Mineral composition and **(b)** proportion of each mineral. Mica and chlorite composed of mica, mica–chlorite mix, chlorite, and feldspars composed of Na/Ca-feldspar and K-feldspar.

derived from different geological sources in the late 1900s compared with the other periods.

## 4.2   Possible causes of mineralogical variation

One of the possible causes of the temporal variations in the silicate mineral composition is a surface temperature change in Greenland. Reconstructions of temperature variability in

Greenland have revealed there were two intense warming periods (1920s–1940s and since the 1990s) and a cooling period (1950s–1980s) in the past 100 years (e.g. Box et al., 2009; Kobashi et al., 2011; Cappelen, 2019). These trends were strong in the western coastal region and were similar to those of the silicate mineral compositions. The proportion of Type A minerals was low in the samples from 1915 to 1949, increased from 1950 to 2004, and decreased again after 2005.

**Table 2.** Relative abundance (%) of silicate mineral groups for each sample.

| Sample period | Kaolinite | Pyrophyllite | Smectite | Illite–smectite | Illite | Mica | Mica–chlorite | Chlorite | Na/Ca-feldspar | K-feldspar | Mafic | Quartz | Unknown |
|---|---|---|---|---|---|---|---|---|---|---|---|---|---|
| 1915–1919 | 17.7 | 15.6 | 0.0 | 6.4 | 8.5 | 1.4 | 9.2 | 5.0 | 7.1 | 6.4 | 5.0 | 14.9 | 2.8 |
| 1920–1924 | 18.8 | 2.9 | 2.2 | 5.1 | 4.3 | 3.6 | 22.5 | 2.9 | 4.3 | 6.5 | 2.2 | 20.3 | 4.3 |
| 1925–1929 | 19.7 | 10.2 | 0.7 | 4.4 | 2.9 | 4.4 | 21.9 | 4.4 | 4.4 | 0.7 | 8.8 | 13.1 | 4.4 |
| 1930–1934 | 19.7 | 15.0 | 2.4 | 5.5 | 0.0 | 4.7 | 25.2 | 4.7 | 3.1 | 0.0 | 0.8 | 13.4 | 5.5 |
| 1935–1939 | 13.7 | 15.3 | 2.4 | 15.3 | 2.4 | 8.1 | 10.5 | 1.6 | 4.0 | 0.8 | 4.8 | 14.5 | 6.5 |
| 1940–1944 | 16.3 | 7.4 | 0.0 | 5.9 | 3.7 | 4.4 | 16.3 | 5.2 | 3.7 | 1.5 | 3.7 | 22.2 | 9.6 |
| 1945–1949 | 21.4 | 7.6 | 0.8 | 10.7 | 3.1 | 1.5 | 13.0 | 6.1 | 14.5 | 3.8 | 4.6 | 9.9 | 3.1 |
| 1950–1954 | 38.8 | 20.9 | 0.7 | 11.2 | 0.0 | 2.2 | 6.7 | 3.7 | 3.0 | 0.0 | 0.7 | 5.2 | 6.7 |
| 1955–1959 | 31.1 | 16.3 | 0.7 | 14.1 | 2.2 | 6.7 | 9.6 | 5.9 | 3.0 | 0.0 | 3.7 | 3.7 | 3.0 |
| 1960–1964 | 37.9 | 21.6 | 0.9 | 12.1 | 2.6 | 0.0 | 8.6 | 0.9 | 3.4 | 0.9 | 0.9 | 3.4 | 6.9 |
| 1965–1969 | 35.0 | 13.3 | 0.0 | 13.3 | 2.8 | 0.7 | 11.9 | 1.4 | 4.2 | 1.4 | 2.1 | 5.6 | 8.4 |
| 1970–1974 | 30.2 | 7.2 | 0.7 | 4.3 | 1.4 | 2.2 | 15.1 | 11.5 | 2.9 | 2.9 | 1.4 | 5.8 | 14.4 |
| 1975–1979 | 58.6 | 12.8 | 0.0 | 6.8 | 3.0 | 1.5 | 3.0 | 0.8 | 3.8 | 0.0 | 0.8 | 5.3 | 3.8 |
| 1980–1984 | 66.7 | 4.9 | 0.0 | 3.5 | 0.7 | 4.2 | 4.2 | 0.7 | 2.8 | 2.1 | 2.1 | 2.8 | 5.6 |
| 1985–1989 | 40.1 | 26.3 | 0.7 | 8.0 | 0.7 | 1.5 | 7.3 | 1.5 | 3.6 | 0.7 | 0.0 | 5.1 | 4.4 |
| 1990–1994 | 39.2 | 20.0 | 0.7 | 5.9 | 0.7 | 0.0 | 2.2 | 4.4 | 10.4 | 0.7 | 2.2 | 8.9 | 4.4 |
| 1995–1999 | 44.5 | 7.3 | 1.5 | 3.6 | 2.9 | 0.7 | 5.1 | 1.5 | 9.5 | 2.9 | 8.0 | 5.1 | 7.3 |
| 2000–2004 | 62.5 | 12.5 | 0.0 | 2.2 | 2.9 | 0.7 | 5.9 | 1.5 | 0.7 | 0.0 | 0.0 | 8.1 | 2.9 |
| 2005–2009 | 4.8 | 5.6 | 1.6 | 3.2 | 10.4 | 2.4 | 20.8 | 8.8 | 12.8 | 7.2 | 3.2 | 10.4 | 8.8 |
| 2010–2013 | 10.1 | 10.1 | 0.0 | 7.1 | 7.1 | 2.0 | 14.1 | 5.1 | 12.1 | 5.1 | 5.1 | 11.1 | 10.1 |

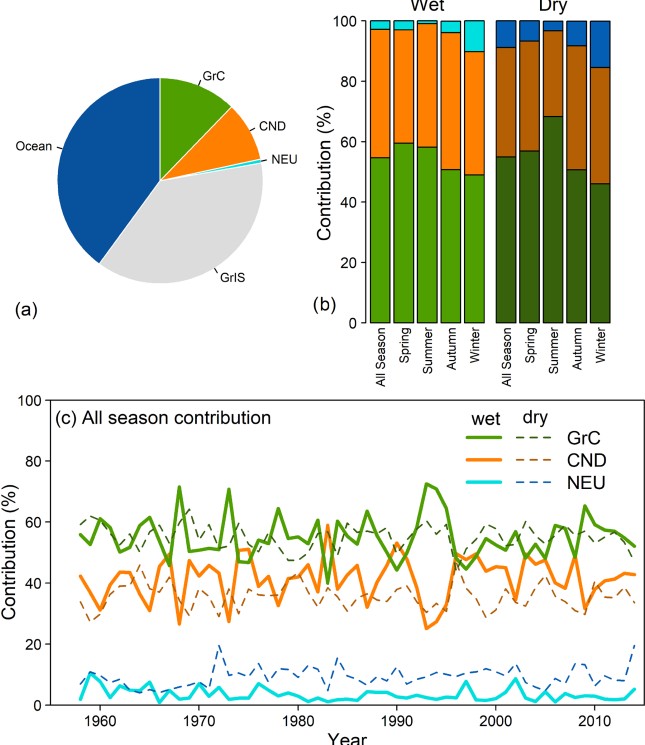

**Figure 8. (a)** Contribution of an air mass from possible source areas from 1958 to 2013, for which precipitation at the Sigma-D site is taken into account. **(b)** Seasonal and **(c)** annual variations in the regional contribution of air mass to the SIGMA-D site through wet and dry depositions excluding the ice sheet and ocean areas. GrC, GrIS, CND, and NEU denote the ice-free Greenland coastal region (Fig. 1), Greenland Ice Sheet, Canada, and northern Eurasia, respectively (Fig. 2a).

In contrast, the proportion of Type B, C, D, and E minerals showed the opposite trend. Therefore, Type A minerals were abundant in the colder periods, whereas Type B, C, D, and E minerals were abundant in the warmer periods (Fig. 9). These results suggest that the multi-decadal variation in SIGMA-D ice core silicates was likely affected by local temperature changes in Greenland.

The North Atlantic Oscillation (NAO) is also thought to be a possible cause of compositional variations in the silicate minerals. The NAO is known to show inter-decadal variations and is strongly related to the incidence and intensity of blocking high pressure over Greenland (Woollings et al., 2010; Hanna et al., 2014). Thus, the NAO can change the atmospheric circulation patterns and transportation processes associated with the ice core dust, which could be related to inter-decadal variations in the silicates. However, there is no clear correlation between the NAO and silicate mineral records. One of the reasons for this may be the low sampling resolution, which makes it difficult to determine a correlation with the NAO index.

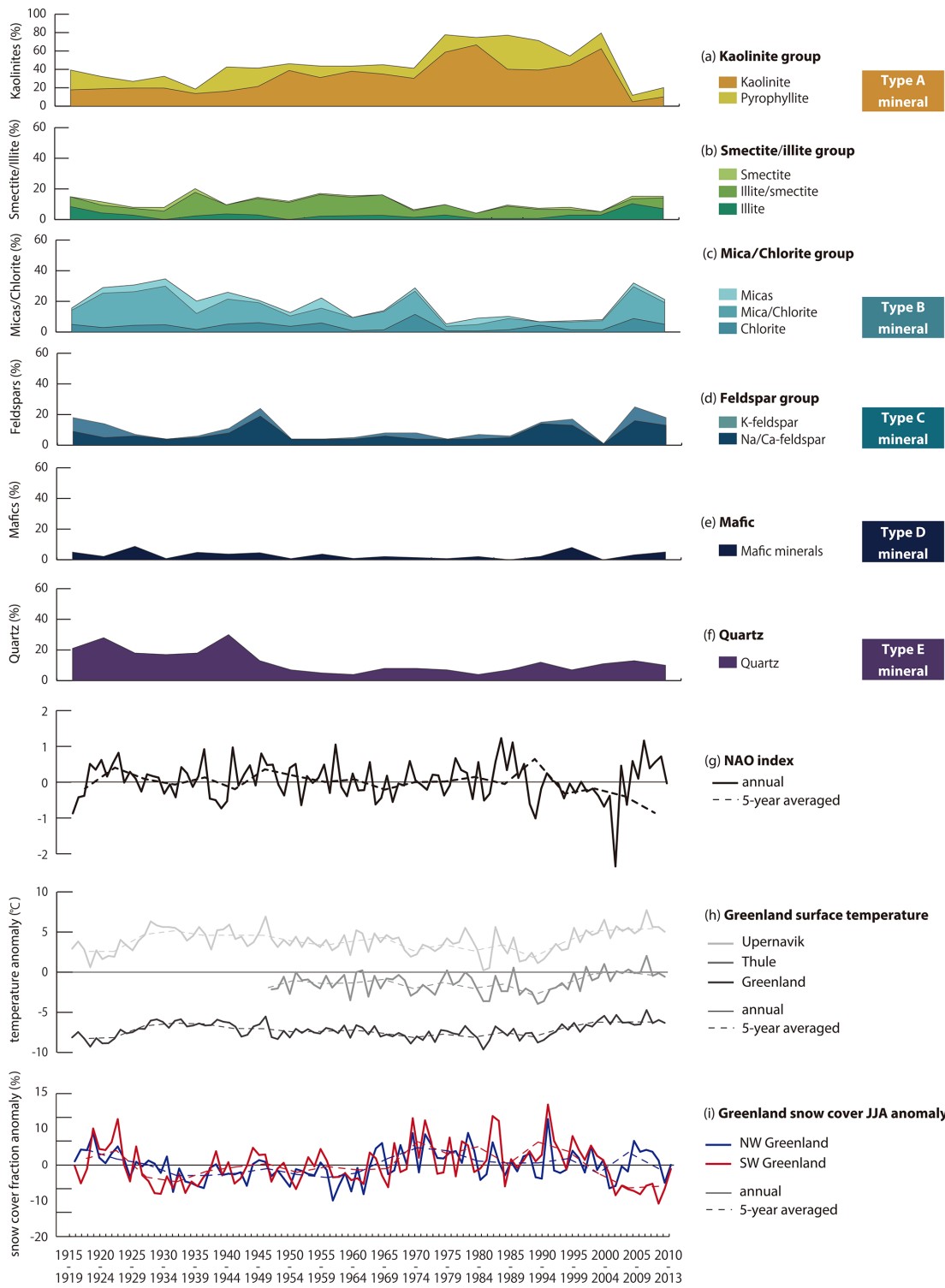

**Figure 9.** Comparison of historical changes in proportion of silicate minerals from the SIGMA-D ice core (**a**: kaolinite group, **b**: smectite group, **c**: mica–chlorite group, **d**: feldspar group, **e**: mafic mineral, **f**: quartz) with those in (**g**) the North Atlantic Oscillation index (NAO; Hurrell and National Center for Atmospheric Research Staff, 2020) and (**h**) surface temperature anomalies and (**i**) snow cover fraction anomalies in Greenland. Surface temperature anomalies deviate from the 1948–2013 average in Thule. The temperature record of Greenland is from Berkeley Earth, and Thule (Pittufik CE1) and Upernavik in western Greenland located 100 km south and 650 km southeast of Qaanaaq are from Cappelen (2019). Snow cover fraction anomalies deviate from the 1915–2013 average in NW and SW Greenland.

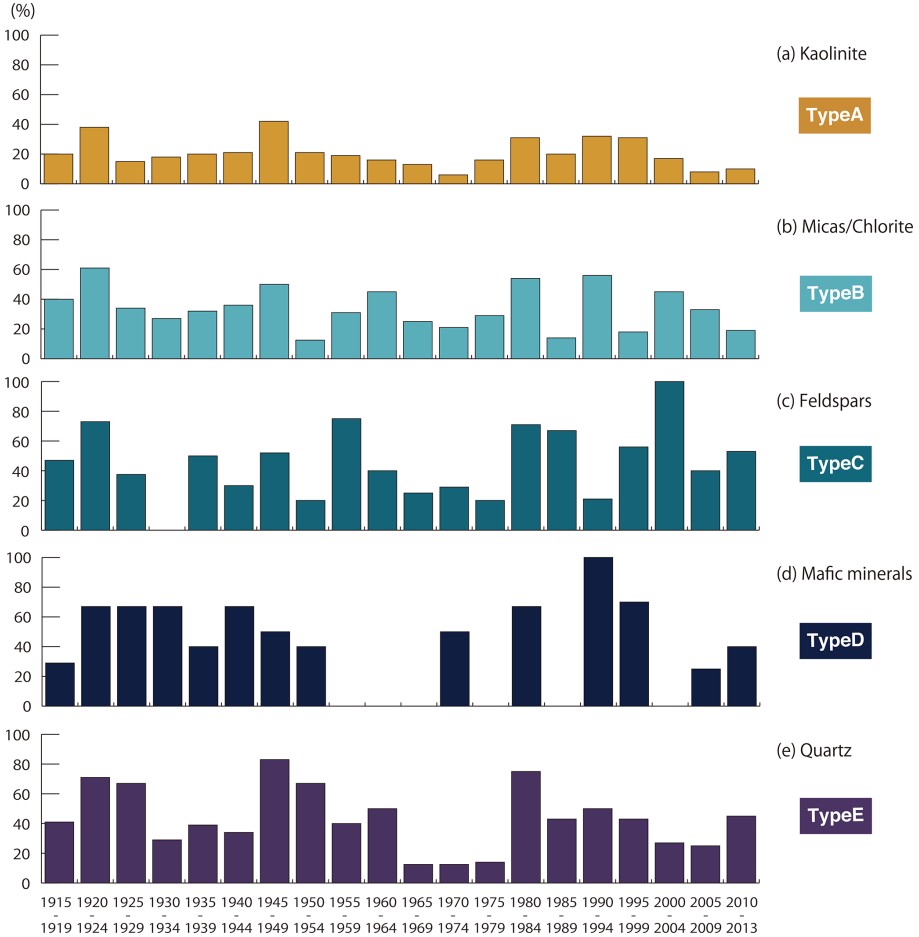

**Figure 10.** Historical changes in proportion of large particles (diameter > 2 µm) in **(a)** kaolinite, **(b)** mica–chlorite, **(c)** feldspars, **(d)** mafic minerals, and **(e)** quartz.

In addition to the NAO, sulfate aerosols originating from volcanic eruptions have also been identified as another important cause of the cooling in Greenland, especially along the western ice sheet margins during the 1900s (Box et al., 2009), such as the Mt. Agung eruption in 1963, the Mt. St. Helens eruption in 1980, and the Mt. Pinatubo eruption in 1991. However, the chemical compositions of ash from these volcanoes (Taylor and Lichte, 1980; Pallister et al., 1992; Devi et al., 2019) were different from the composition of the Type A minerals that were abundant in the cooling period. The SEM observations also did not identify ice core minerals exhibiting morphological characteristics of volcanic ash. These results indicate that the effect of volcanic materials on the variation in silicate mineralogy may be negligible in the SIGMA-D ice core.

### 4.3 Possible sources for mineral dust in the SIGMA-D ice core

The trajectory analysis revealed that the majority of the air mass came from the western coast of Greenland and that a smaller proportion came from northern Canada between 1958 and 2014. The contribution from these two possible source regions showed little inter-annual variabilities (Fig. 8b and c), indicating that the transportation processes of the ice core dust have not substantially changed on an annual basis over the last 5 decades. Thus, the variations in the geological origins of the ice core dust were unlikely due to changes in air mass transportation. An alternative cause of dust variability could be a change in the surface conditions of the source areas. Retreats of the ice sheet and local glaciers have accelerated since 2000 in Greenland, increasing the exposure of the ground surface in snow/ice-covered areas in the coastal region (e.g. van den Broeke et al., 2009; Bendixen et al., 2017). Furthermore, the modelled snow cover fraction anomaly during summer (June, July, and August) on the northwest and southwest coasts of Greenland is negatively consistent with the temperature anomalies (Fig. 9h and i). The snow cover fractions are lower during the two warming periods of 1925–1960 and 1990–2013. Given that the snow cover fraction should directly relate to the snow cover duration, this result suggests that the snow cover duration on

the west coast of Greenland was shortened in the warming periods and thus might also have contributed to the increase in local dust emissions. Bullard and Mockford (2018) analysed records of dust events in the western Greenland coastal region and revealed that the annual severity of dust emissions was higher from 2000 to 2010 than during the preceding decades. This was likely due to increasing meltwater runoff delivering sediments from the ice sheet to outwash plains with the increase in atmospheric temperature. High dust emissions on the western coast of Greenland occurred in spring and summer, when the snow cover is rapidly decreasing (Bullard and Mockford, 2018). Our trajectory analysis also indicated that the air mass contribution from the Greenland coast was slightly larger in spring and summer than in autumn and winter (Fig. 8b). Therefore, the snow/ice cover duration in the Greenland coastal region was shortened by the recent warming during the melt season, causing an increase in the local supply of dust to the SIGMA-D site. Although no satellite observations are available for the first warming period (1920s–1940s), aerial photos, maps, and paintings indicated ice retreat in Greenland (Box and Herrington, 2007). Thus, the abundant Type B, C, D, and E minerals found in the two warming periods were likely due to an increase in dust sourced from local ice-free areas.

The ice core dust morphology and composition also indicated a contribution from local sediment in the warming periods. Simonsen[TS11] et al. (2019) used particles with diameters of < 2 and > 8 µm as indicators of distant and local dust sources, respectively, for the Renland Ice Cap Project ice core in eastern Greenland. The size distribution of SIGMA-D ice core dust showed that the samples from the cold period (1960–1989) contained fewer particles with diameters of > 8 µm (0–2 particles) than those from the warm periods (1915–1959 and 1990–2013; 1–9 particles). Furthermore, the higher proportions of Type C and E minerals in the SIGMA-D ice core samples in 1915–1949 and 1990–2013 correspond to higher concentrations of the two minerals in the surface dust and soil on and around the Qaanaaq Glacier (Nagatsuka et al., 2014). Since the Abakus laser particle sensor connected to the CFA system detected particles with diameters of 1.5 and 15 µm and CFA analysis has not been performed since 2003 (above a depth of 6.35 m) due to poor core quality, we cannot compare the size distribution data with that measured by the SEM. However, the Abakus dust profiles (Fig. A3) showed higher concentration for the particles with diameters of > 1.5, > 5, and > 8 µm in the warm period (1915 to the 1950s) compared with the cold period (1950–2000, except for a large peak in 1978), which supports an increase in local dust contribution from 1915 to the 1950s as shown by the SEM–EDS results.

Previous studies indicated dust transport from distant deserts, such as those in Asia and Africa, which are another possible source for three of the mineral types (B, C, and E) found at high-elevation sites on the Greenland Ice Sheet during the past century (Bory et al., 2003a; Drab et al., 2002).

However, our trajectory analysis showed little contribution from these regions in the 7 d back trajectory. A similar analysis of an ice core on the southeastern coast of Greenland suggested that air mass contribution from Asian and arid regions was negligible, even for the 25 d back trajectory (Iizuka et al., 2018). Schüpbach et al. (2018) also reported little contribution of air mass from Asian arid regions and addressed a limitation of the back-trajectory analysis, namely that it could not capture dust transport from Asia to Greenland, which might be through the upper troposphere. Bory et al. (2003b[TS12]) suggested that sources providing dust to an ice core site are dependent on distance from the ice sheet margin and/or the altitude and that long-range transport from Asian deserts likely accounts for most of the dust deposited at interior sites (NGRIP, GRIP, Site-A, and Dye-3), whereas local sources represent an additional and primary contributor at coastal sites in Greenland (Hans Tausen and Renland). Our 7 d back-trajectory analysis shows the significantly low contribution of air masses from Asian and arid regions (less than 0.1 %), whereas the air mass contribution from the Greenland coast is high (7 %–14 %), especially for Renland and Hans Tausen, followed by the Sigma-D site (Fig. 11a and d). The mineralogical composition of SIGMA-D also showed a substantially lower proportion of Type B (illite) and Type E minerals and a higher proportion of Type A minerals (6 %–18 %, 3 %–22 %, 5 %–63 %) compared with other Greenland ice core dust originating from Asian deserts (e.g. GRIP: 23 %–35 %, 31 %–48 %, 4 %–16 %; Svensson et al., 2000). Thus, very little of the mineral dust in the ice core from SIGMA-D, which is located in a coastal area, may have come from distant deserts, whereas a large proportion likely came from the local areas in the warming periods.

Possible sources of ice core dust in the colder period (1950–2004) are likely to be found in low to mid-latitudes because the Type A mineral that is typical of humid tropical climatic zones such as modern-day Africa, South America, and Southeast Asia was abundant in that period. Although back-trajectory analysis cannot estimate contributions from distant sources as described above, it is unlikely that large amounts of ice core dust were transported from such tropical regions. However, our back-trajectory analysis suggests that northern Canada might also be a possible source of Type A minerals, and some studies support this argument (Fig. 11b). For example, Darby (1975) analysed the clay mineral composition of marine sediments from deep-sea cores in the Arctic Ocean and revealed that there was abundant kaolinite (Type A minerals) apparently derived from shale and soils of northern Alaska and northern Canada, which were relict deposits of warmer climates in the Tertiary. This abundant kaolinite was deposited in a non-marine environment (Allen and Johns, 1960). Clay mineralogy of North Sea basin sediment cores also revealed that increased kaolinite concentrations were associated with the Paleocene–Eocene Thermal Maximum (Kemp et al., 2016). The Type A minerals were likely transported from such ancient soils formed by chemi-

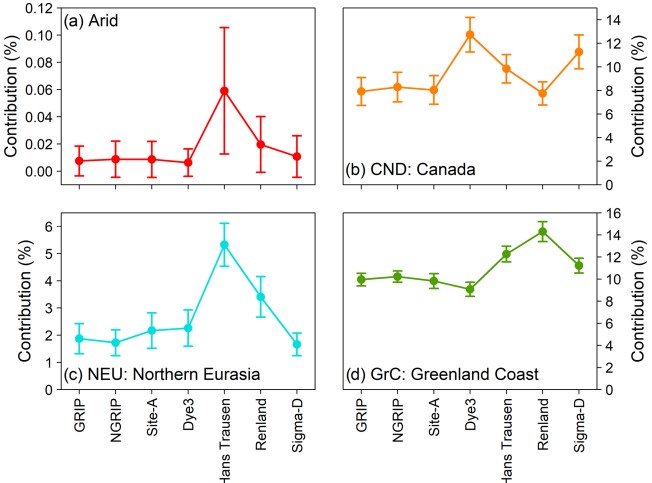

**Figure 11.** Air mass contribution for the 7 d back trajectory from **(a)** arid regions (China, Central Asia, and the Middle East), **(b)** North America (Canada and US), **(c)** Eurasia (EU and Russia), and **(d)** the Greenland coast at the seven ice core sites on Greenland. Error bars indicate the standard deviation of contributions from 1981 to 2010.

cal weathering in high-latitude areas in past warming events. Thus, we concluded that northern Canada is likely the best candidate for the SIGMA-D ice core dust source during the cold period.

The SEM–EDS results of the SIGMA-D ice core dust can demonstrate variations in the ice core dust sources over the past 100 years. The relatively higher proportion of Type A minerals (more than 40 %) during these periods indicates that the ice core dust was constantly supplied from a distant source (mainly northern Canada) to the SIGMA-D site and that the source areas have not changed over the past 100 years. However, dust was additionally provided from local ice-free areas in the warm periods (1915–1950 and 2005–2013) because the snow/ice cover duration in the Greenland coastal region was shortened by the recent warming during the melt season.

## 5   Conclusions

Analysis of the SEM–EDS of individual dust morphology and mineralogy in the SIGMA-D ice core revealed that the ice core dust consisted mainly of silicate minerals, including quartz, feldspars, and mafic minerals, and clay minerals, including kaolinite, illite, smectite, mica, and chlorite. Most of the particles had a diameter of $< 2\,\mu$m, implying that the ice core contained mainly long-range-transported wind-blown mineral dust. The silicate mineral composition varied substantially on multi-decadal and inter-decadal scales. The multi-decadal variation trend differed among mineral types formed in different source areas, which corresponded to surface temperature changes in Greenland; kaolinite, which

is typical of humid tropic climatic zones, was abundant in the colder period (1950–2000), whereas mica, chlorite, feldspars, mafic minerals, and quartz, which are generally formed in arid, high-latitude, and/or local areas, were abundant in the warmer periods (1915–1949 and 2005–2013). This indicates that the ice core minerals originated from different geological sources in different periods during the past 100 years. The multi-decadal variation in the relative abundance of the minerals was likely affected by local temperature changes in Greenland. The trajectory analysis showed that the air mass arriving at the SIGMA-D site came mainly from the western coast of Greenland and that a smaller proportion came from northern Canada during 1958–2013. The contributions from the two showed little inter-annual variability, indicating that an alternative cause of variability in the geological origins of the ice core dust was likely to be a change in the surface conditions of source areas rather than in air mass transportation. The abundant mineral types in the two warmer periods might be explained as an increase in dust sourced from local ice-free areas resulting from shortened snow/ice cover duration in the Greenland coastal region caused by the recent warming during the melt season. Meanwhile, ancient deposits in northern Canada, which were formed in past warmer climates, seem to be the best candidate during the colder period (1950–2004). We concluded that ice core dust was constantly supplied from distant sources (mainly northern Canada) to the SIGMA-D site as well as local ice-free areas in the warm periods. Although further analyses are needed to identify the cause of inter-decadal variations in ice core dust, our study is the first to demonstrate a high-temporal-resolution record of mineral composition in a Greenland ice core over the past 100 years.

## Appendix A

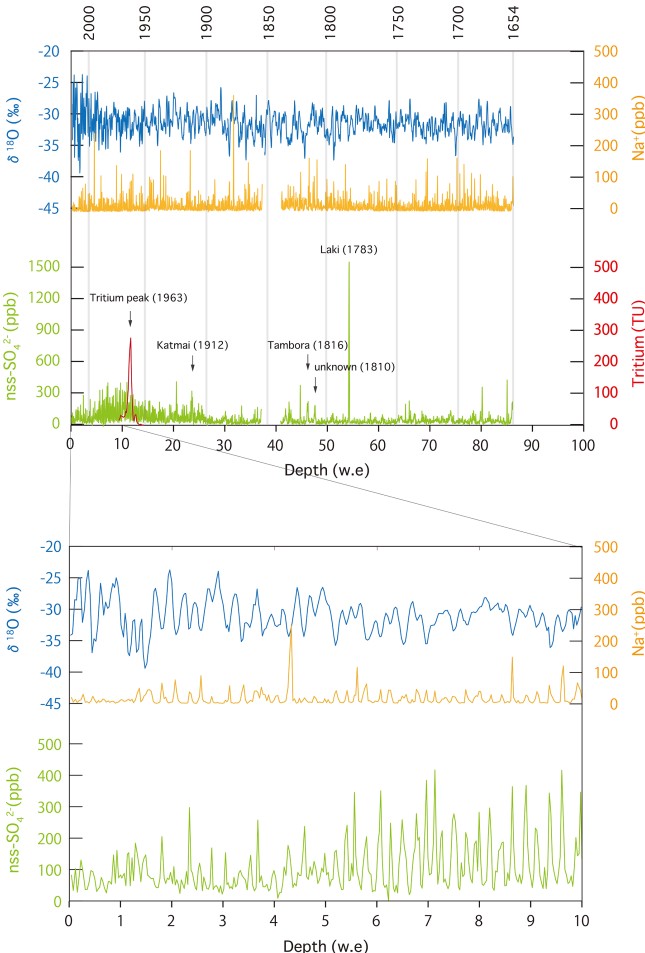

**Figure A1.** $^{18}$O, Na$^+$, nssSO$_4^{2-}$, and tritium records in the upper 112.87 m (86.06 m w.e.) of the SIGMA-D ice core. Major volcanic signals we identified are shown in the SO$_4^2$ record. The bottom plots show the enlarged record from 0 to 10 m w.e.

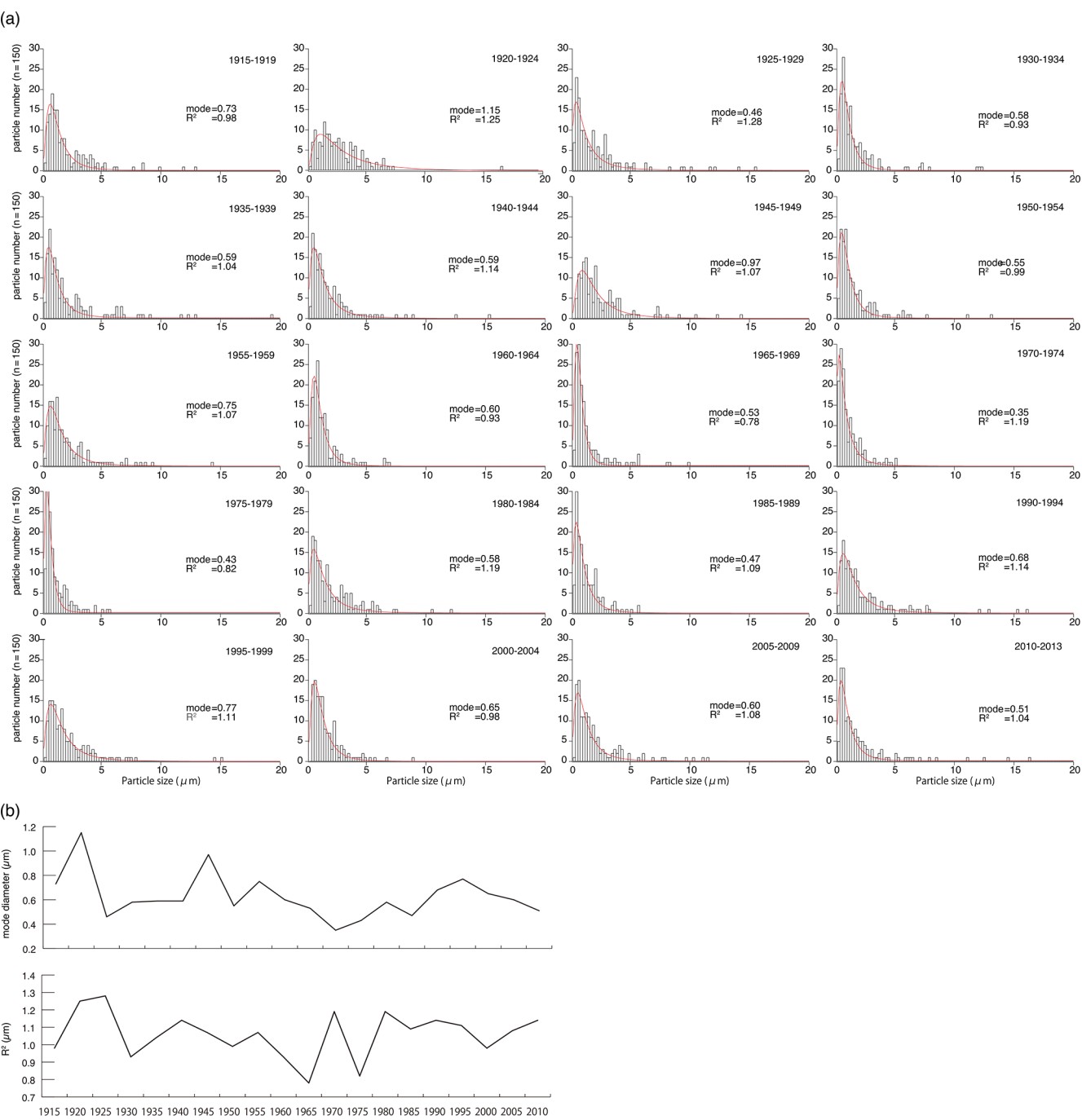

**Figure A2. (a)** Comparison of particle size distribution and log-normal fitting results (mode: mode diameter and $R^2$: half peak width) of the ice core minerals among the samples. **(b)** Historical changes in the mode diameter and $R^2$.

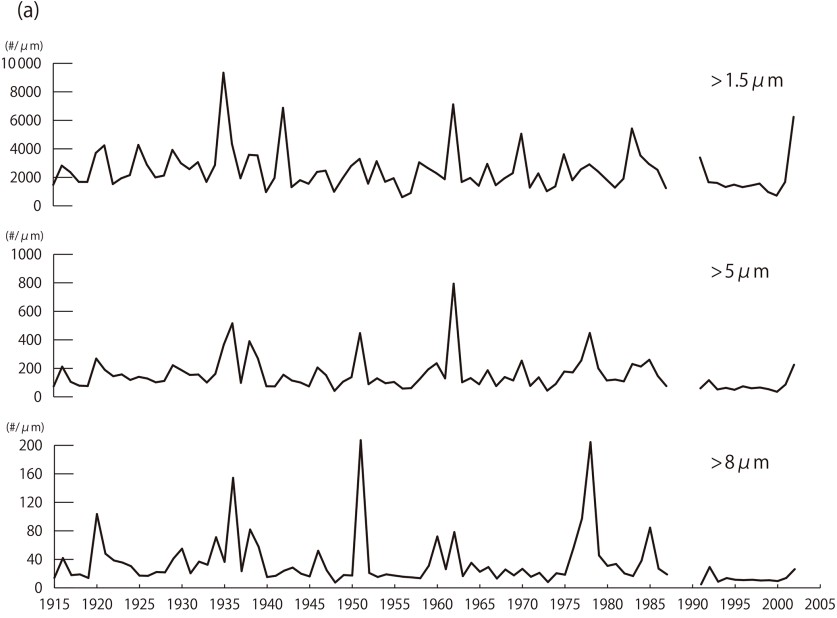

**Figure A3. (a)** Annual average number concentration of the SIGMA-D ice core dust particles with diameters of > 1.5, > 5, and > 8 µm from 1915 to 2002 as measured by the Abakus laser particle sensor and **(b)** the averaged concentrations from 1915 to 1954 and from 1955 to 1959.

**Data availability.** Datasets used in this study are available at the following DOIs.

MIROC6 model output prepared for CMIP6 LS3MIP experiments is available at

https://doi.org/10.22033/ESGF/CMIP6.5622 (land-hist: Onuma and Kim, 2020a),

https://doi.org/10.22033/ESGF/CMIP6.5627 (land-hist-cruNcep: Onuma and Kim, 2020b),

https://doi.org/10.22033/ESGF/CMIP6.5628 (land-hist-princeton: Onuma and Kim, 2020c), and

https://doi.org/10.22033/ESGF/CMIP6.5629 (land-hist-wfdei: Onuma and Kim, 2020d).

Data on $\delta^{18}$O and concentrations of sodium and sulfate ions ($Na^+$ and $SO_4^{2-}$) will be submitted to the ADS (Arctic Data archive System) database TS13 for public use in further analysis.

**Author contributions.** NN designed the study and carried out the ice core dust analysis and wrote the manuscript with the help of KGA and KF. KF, SM, YO, YK, MM, and HM drilled the ice core. AT, SM, and MK obtained ion concentration and water isotope data. NN, AT, KF, SM, and MK analysed the chronology of the ice core. KF conducted the back-trajectory analysis. YO conducted the CMIP6 model analysis. RD, MH, JO, YOT, KK, and KGA conducted the CFA analysis. KGA and YOT analysed the dust data. TA initiated the project. All authors discussed and commented on the paper.

**Competing interests.** The authors declare they have no conflicts of interest.

**Acknowledgements.** We would like to thank Tetsuhide Yamasaki for general fieldwork support. This research was funded by a Grant-in-Aid for JSPS Fellows, by JSPS KAKENHI (grant numbers 23221004 and 16H01772 (SIGMA project) and 15H01731, 15K16120, 16J08380, 16H06291, 18H03363, 18H04140, 19K20443, and 20H04980). This work was supported by the Integrated Research Program for Advancing Climate Models (TOUGOU) grant number JPMXD0717935457 from the Ministry of Education, Culture, Sports, Science and Technology (MEXT), Japan. This study was also supported by the Arctic Challenge for Sustainability (ArCS) (grant number JPMXD130000000), the Arctic Challenge for Sustainability II (ArCS II) (grant number JPMXD1420318865), the Environment Research and Technology Development Fund of the Environmental Restoration and Conservation Agency of Japan (grant numbers JPMEERF20172003 and JPMEERF20202003), and a grant for the National Institute of Polar Research, Japan (grant number KP305).

**Financial support.** This research has been supported by the Japan Society for the Promotion of Science (JSPS) Fellowship (SIGMA Project (grant nos. 23221004 and 16H01772) and 15H01731, 15K16120, 16J08380, 16H06291, 18H03363, 18H04140, 19K20443, 20H04980) TS14.

**Review statement.** This paper was edited by Elizabeth Thomas and reviewed by Anders Svensson and Laluraj C. M.

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

Please note the remarks at the end of the manuscript.

**Remarks from the language copy-editor**

CE1 Do you mean Pituffik? Please check.

**Remarks from the typesetter**

TS1 Please confirm the corrected citation.

TS2 Please note that the year has been corrected in accordance with the one cited in the reference list. Please check and confirm.

TS3 Please confirm the corrected citation.

TS4 Please provide the short title.

TS5 Please confirm the corrected citation.

TS6 Please note that the year has been corrected in accordance with the one cited in the reference list. Please check and confirm.

TS7 Please confirm the corrected citation.

TS8 Please confirm the corrected citation.

TS9 The composition of Figs. 2, 4, 8, 11, and A2–A3 has been adjusted to our standards.

TS10 Please confirm the corrected name.

TS11 Please confirm the corrected name.

TS12 Please note that the year has been corrected in accordance with the one cited in the reference list. Please check and confirm.

TS13 Is it possible to provide the homepage/URL.

TS14 Please note that there is funding information given in the acknowledgements, but you did not indicate any funding upon manuscript registration. Therefore, we were not able to complete the financial support statement. This section has to be written in neutral language, meaning that there should be no expression of thanks or gratitude (which should be in the acknowledgements) and the format for this section is as follows: "This research has been supported by the *name of institute, organization, etc.* (grant no. *number* (if available)) and the *name of institute, organization, etc.* (grant no. *number*)". Please provide the missing information and double-check your acknowledgements to see whether repeated information can be removed from the acknowledgements. Thanks.

TS15 Please provide the publisher/publishing institution.

TS16 Please confirm the name of the journal.

TS17 Please confirm the name of the journal.

TS18 Please provide this information.

TS19 Please cite the provider/repository.

TS20 Please provide this information.

TS21 Please note the corrected year.

TS22 Please confirm the corrected reference.