# Peer review of "Variations in mineralogy of dust in an ice core obtained from northwestern Greenland over the past 100 years"

_Climate of the Past, 2020_

## Referee Comment (RC1) · Anders Svensson (Referee) · 26 Dec 2020

The manuscript is concerned with a century long dust record obtained from the northwestern Greenland SIGMA-D ice core. The dust is characterized in terms of mineralogy (SEM+EDS), size and roundness at 5-year resolution. Furthermore, a trajectory analysis is performed in order to track the origin of the dust. It is found that the dust is a mixture of local dust and dust from more distant sources and that the fraction of local dust varies with coastal weather and snow cover.

General comments

The manuscript is generally making a good case describing the dust variability in North-western Greenland of the last century. To investigate the dust source, it would however be very helpful to have a few isotopic analyses of Sr or Nd that are probably more indicative of the dust origin than the mineralogy and certainly more informative than eg the roundness parameter. The argumentation and thus the conclusions concerning the dust source being at high, mid or low latitudes based on mineralogy are in my view somewhat weak without the isotopes. Likewise, the lack of isotopic analyses makes comparison to similar analyses from many other Greenland ice cores impossible. Indeed, it would be a surprise if we have substantial amounts of low latitude dust deposited in Northern Greenland today.

I have my doubts about the trajectory analysis suggesting that none of the dust in the ice core originates from Asia. Large Asian dust clouds are observed on satellite images following the dominating westerly wind pattern. During the last glacial the majority of the Greenland dust had this source. Likewise, several tephra studies show that volcanic material of high-latitude Asian and Alaskan origin makes it to Greenland, whereas no tephra from Europe or lower latitudes ever made it to Greenland to my knowledge. Today the major Greenland dust source is less well-known and a more local dust contribution certainly is a possibility, but I would suspect a fraction of the dust still to originate in Asia today. Without the isotopic tracers, however, it is difficult to know, and trajectory analysis may not tell the complete story. Please compare your trajectory analysis to that made in Schüpbach et al., 2018, and comment on differences in methodology and results. In particular, Schüpbach et al., 2018, supplementary figure 1 may be relevant.

In Simonsen et al., 2019 (full reference below), the dust size distribution (or the contribution of large fraction particles) is seen as a strong indicator of local versus distant dust sources. Large particles cannot be transported over long distances, so the presence of large particles in the ice core is attributed to local dust sources. Does the dust size distributions in your study support your conclusions of local versus long-range dust

source variations?

I find the manuscript is generally well written with good figures and referencing, except the introduction that I think needs a rework both language-wise and concerning references. The cited papers appear not to be up to date; below I give some additional suggestions for recent work that I think should be referred to and possibly commented on.

Specific comments:

Mention the name of the analysed ice core in the title or in the abstract.

It would be relevant to cite this recent paper in the introduction, since the idea of more local dust in Greenland ice cores is also discussed here although on a different time scale:

Simonsen, M. F., Baccolo, G., Blunier, T., Borunda, A., Delmonte, B., Frei, R., Goldstein, S., Grinsted, A., Kjær, H. A., Sowers, T., Svensson, A., Vinther, B., Vladimirova, D., Winckler, G., Winstrup, M., and Vallelonga, P.: East Greenland ice core dust record reveals timing of Greenland ice sheet advance and retreat, Nature Communications, 10, 10.1038/s41467-019-12546-2, 2019.

l. 19.: 'changed periodically' leaves the impression that the dust composition varies with a certain periodicity, ie with a certain frequency. Probably the authors mean to say that the dust had different composition in different periods?

l. 23.: 'This indicates. . .' -> 'Comparison to Greenland surface temperature records indicates. . .'

l. 24.: 'The trajectory . . .' -> 'A trajectory. . .' or 'Trajectory. . .'

l. 30.: 'Past dust composition. . .' -> 'Past ice-core dust composition records have revealed substantial variations in the concentration. . .' It is unclear which time period you refer to here. Is this about glacial cycles (Antarctica), the last glacial cycle, the

Holocene, historical times, or on a seasonal time scale? With this postulate you need to cite a reference.

l. 31-32.: 'climate signals'. Are you referring to ice core water isotopes? 'atmospheric circulation', what is referred to here? Again the time period is not specified. Again you will need to cite published work for this statement.

l. 29-40.: I think you also need to cite at least some of those classic studies:

Ruth, U., Wagenbach, D., Steffensen, J. P., and Bigler, M.: Continuous record of microparticle concentration and size distribution in the central Greenland NGRIP ice core during the last glacial period, Journal of Geophysical Research, 108, 10.1029/2002jd002376, 2003.

Petit, J. R., Mounier, L., Jouzel, J., Korotkevich, Y. S., Kotlyakov, V. I., and Lorius, C.: Palaeoclimatological and chronological implications of the Vostok core dust record, Nature, 343, 56-58, 1990.

Lambert, F., Delmonte, B., Petit, J. R., Bigler, M., Kaufmann, P. R., Hutterli, M. A., Stocker, T. F., Ruth, U., Steffensen, J. P., and Maggi, V.: Dust-climate couplings over the past 800,000 years from the EPICA Dome C ice core, Nature, 452, 616-619, 2008.

l. 42.: What is meant by 'rarely'? Do you have evidence of changing isotopic rations during transport? If so, please cite relevant reference.

l. 41-50.: You probably should cite some of the more recent papers as well:

Újvári, G., Stevens, T., Svensson, A., Klötzli, U. S., Manning, C., Németh, T., Kovács, J., Sweeney, M. R., Gocke, M., Wiesenberg, G. L. B., Markovic, S. B., and Zech, M.: Two possible source regions for central Greenland last glacial dust, Geophysical Research Letters, 42, 10399-10408, 10.1002/2015GL066153, 2015.

Han, C., S. D. Hur, Y. Han, K. Lee, S. Hong, T. Erhardt, H. Fischer, A. M. Svensson, J. P. Steffensen, P. Vallelonga. High-resolution isotopic evidence for a potential Saharan provenance of Greenland glacial dust. Scientific Reports 8:15582 | DOI:10.1038/s41598-018-33859-0 3, 2018. l. 51-60.:

Probably you should cite this paper:

Obbard, R. W., Baker, I., and Prior, D. J. (2011). Instruments and Methods A scanning electron microscope technique for identifying the mineralogy of dust in ice cores. Journal of Glaciology 57, 511–514.

l. 63.: '. . . in the ice sheet' -> '. . . to the ice sheet'.

l. 61-72.: This recent paper would also be relevant for the introduction:

Zhang, P., Jeong, J.-H., Yoon, J.-H., Kim, H., Wang, S.-Y. S., Linderholm, H. W., Fang, K., Wu, X., and Chen, D.: Abrupt shift to hotter and drier climate over inner East Asia beyond the tipping point, Science, 370, 1095-1099, 10.1126/science.abb3368.

l. 179.: 'Dating of the SIGMA-D ice core, which was performed by annual layer counting of $\delta$18O and Na+, showing obvious seasonal variations (Fig. A1).' -> 'Dating of the SIGMA-D ice core was performed by annual layer counting of $\delta$18O and Na+ that show obvious seasonal variations (Fig. A1).'

l. 246.: 'cyclically' suggests a periodicity or frequency. What is the period or frequency on two different time scales referred to?

l. 250.: Are you suggesting a major contribution of low to mid-latitude dust to Northern Greenland for the period 1950-2004? What is the potential source area and why should it be turned on in 1950 and off in 2004?

l. 309.: 'Previous studies. . .' What are those previous studies and are they concerned with dust transport to Greenland in the last century?

---

## Referee Comment (RC2) · Laluraj C.M (Referee) · 4 Jan 2021

Naoko Nagatsuka et al use the upper 38.6 m dust mineral record (1915-2013 CE) of an ice-core of a total 222.72 m (1660-2013 CE) to study the variation in the mineralogy of dust from northwestern Greenland. The authors used 5 year averaged dust mineralogy record (1915-2013 CE) analysed in SEM-EDS instrument from the different sections (5-year resolution) of the firn part of the ice core. The variation in dust mineralogy record was interpreted in terms of their different geological origin with the help of air parcel trajectories during the period 1915-2013 CE. The authors suggested the possibility of dust sources from local ice-free areas during warmer periods and from northern America during colder periods using air trajectories and abundance of kaolinite mineral respectively.

**General comments**

The reporting of new dust mineral record observations in Greenland as well as attempts to identify the sources of dust during warmer/colder periods during 1915-2013 is worthy of publication. However, the report is not suitable for publication as it is. The strength of this work resides only in reporting a mineralogy record of a century time scale from Greenland; there have been no attempts to offer climatic reasons for recent changes in dust mineralogy/sources/production in source/deposition regions. Conversely, perhaps in the attempt to cover so much, the paper fails to go in-depth in a number of aspects of these important subjects and offer no insightful reasons for the possible causes in variation in dust mineralogy and sources changes during the recent colder/warmer periods of last century. In particular, the authors omitted the potential methods (stable isotope ratio of Sr and Nd) to identify the sources of dust in different time scales as well as failing to consider high-resolution fractional dust estimates using water-insoluble particle counting (Coulter counter and laser-sensing particle detector) along with the estimated mineralogy data for dust sourcing/variation of dust mineralogy for strengthening the present argument. Further, lack of trajectories observed from well-known dust sources reported in Greenland (eg:-North Africa/Middle East, and Eastern Asia eg;- Simonsen et al., 2019 and publication by Svensson & group), thereby doubting the uncertainties in using reanalysis data prior to 1979 CE. Additionally, I strongly encourage the authors to consider the full ice core mineralogy/dust records of 223 m or/at least 113 m (1660-2013 CE) for a better understating of variation in mineralogy records in Greenland during a longer period instead of considering the present firn part of the ice core (38 m) where most of the uncertainties are present.

I believe that the above-mentioned issues can be addressed with reasonable additions and extra analysis. I leave it to the Editor to decide whether to recommend, to modify/drop one of the subject study (mineralogy variation in Greenland during the last century) or leave them all.

**Specific comments**

Line No. 23 - This indicates that the multi-decadal variation of the relative abundance of the minerals can be attributed to the local temperature changes in Greenland. The statement is quite arbitrary so may be re-written or removed.

Maybe a statement on the overall implication of your study or future perspectives will be useful at the end of the abstract.

The first paragraph of the introduction may be rewritten or adding a new paragraph by including recent studies (eg:- Zhang et al., 2020;  Bory et al., 2002 & 2003) on dust sources dust in Green land/mineralogy of dust in Greenland/implications in terms of climate changes may be included.

Line No. 103- How reliable the SEM analysis is to estimate the particle size and the number of particle counting from ice core used in the present study? May be compared/clarified here using the different particle count estimate from other instruments.

Line No. 180- Since there are no other non-sea salt sulphate spikes of volcanic origin in the present time scale (1915-2013), Fig. A1 may be included with tritium spike data of chronology as a reference point

Line 245- Is the abundance of silicate minerals due to volcanic ash? There are several volcanic dusts reported in Greenland ice cores during the period between 1915-2013. Please include a discussion on it.

Line No.310:-The choice of 07 days is not explained either and why the author considered only wet deposition?. At least this should be discussed. Trajectories also can never be discussed without cross-checking the results with the general synoptic situation(s) during the transport. Trajectories with links, as shown in Fig. 2, are very unlikely to represent the real path of the air/dust particle.  What does it mean by a little contribution from Asia? please quantify it.

References

Bory, A. J.-M., Biscaye, P. E., Piotrowski, A. M. & Steffensen, J. P. Regional variability of ice core dust composition and provenance in Greenland. Geochem. Geophys. Geosys. 4, https://doi.org/10.1029/2003GC000627 (2003).

Bory, A. J. M., Biscaye, P. E., Svensson, A. & Grousset, F. E. Seasonal variability in the origin of recent atmospheric mineral dust at NorthGRIP, Greenland. Earth Planet. Sci. Lett. 196, 123–134 (2002).

Schüpbach, S., Fischer, H., Bigler, M. et al. Greenland records of aerosol source and atmospheric lifetime changes from the Eemian to the Holocene. Nat Commun 9, 1476 (2018). https://doi.org/10.1038/s41467-018-03924-3

Zhang, P., Jeong, J.-H., Yoon, J.-H., Kim, H., Wang, S.-Y. S., Linderholm, H. W., Fang,K., Wu, X., and Chen, D.: Abrupt shift to hotter and drier climate over inner East Asia beyond the tipping point, Science, 370, 1095-1099, 10.1126/science.abb3368.

---

## Author Comment (AC1) · 27 Jan 2021

We appreciate all the reviewers for their comments on our manuscript. Please, find the replies to the reviewer comments in the pdf file attached.

Please also note the supplement to this comment:
https://cp.copernicus.org/preprints/cp-2020-146/cp-2020-146-AC1-supplement.pdf

---

## Author Comment (AC2) · 27 Jan 2021

We appreciate all the reviewers for taking the time to read our paper and many helpful suggestions. We consider the comments very carefully and will revise the manuscript. Below we provide point-to-point answers to the comments of the two reviewers.

**Dear Dr. Anders Svensson as Reviewer #1**

**[Reviewer comment]**

General comment: To investigate the dust source, it would however be very helpful to have a few isotopic analyses of Sr or Nd that are probably more indicative of the dust origin than the mineralogy and certainly more informative than eg the roundness parameter. The argumentation and thus the conclusions concerning the dust source being at high, mid or low latitudes based on mineralogy are in my view somewhat weak without the isotopes. Likewise, the lack of isotopic analyses makes comparison to similar analyses from many other Greenland ice cores impossible. Indeed, it would be a surprise if we have substantial amounts of low latitude dust deposited in Northern Greenland today. [Author reply]

We agree on the reviewer's comment that the Sr and Nd isotopes are one of the most useful tools to identify mineral dust sources. However, it is difficult to apply the isotope analyses for ice cores with low dust concentration because they need large amounts of dust samples. Indeed, previous studies on the isotope analyses have mostly targeted ice core dust from glacial periods characterized by high dust concentrations (e.g. Biscaye et al., 1997; Svensson et al., 2000). Although some studies analyzed the Sr and Nd isotopes of ice core dust from Holocene when dust concentrations are low, they needed to concentrate decades to thousand years of ice for each sample (e,g, Bory et al., 2003; Han et al., 2018; Simonsen et al 2019) Thus, we think the SEM-EDS analysis is a useful tool to provide a high-temporal-resolution record of composition and sources of ice core minerals during low dust concentration periods.

[Reviewer comment] General comment: I have my doubts about the trajectory analysis suggesting that none of the dust in the ice core originates from Asia. Large Asian dust clouds are observed on satellite images following the dominating westerly wind pattern. During the last glacial the majority of the Greenland dust had this source. Likewise, several tephra studies show that volcanic material of high-latitude Asian and Alaskan origin makes it to Greenland, whereas no tephra from Europe or lower latitudes ever made it to Greenland to my knowledge. Today the major Greenland dust source is less well-known and a more local dust contribution certainly is a possibility, but I would suspect a fraction of the dust still to originate in Asia today. Without the isotopic tracers, however, it is difficult to know, and trajectory analysis may not tell the complete story. Please compare your trajectory analysis to that made in Schüpbach et al., 2018, and comment on

differences in methodology and results. In particular, Schüpbach et al., 2018, supplementary figure 1 may be relevant.

[Author reply]

The model (NOAA HYSPLIT) and datasets (ERA-Interim precipitation) used in our trajectory analysis are same as those of Schüpbach et al. (2018), but the initial height of air mass (50, 500, 1000, 1500 m a.g.l. vs. 100m a.g.l.), and the output figure descriptions (cumulative probability vs. endpoint) are different. We analyzed the 10-day back trajectory from the NEEM and SIGMA-D sites in the same setting for 1979-2014 (4 initial heights), and depicted the same trajectory map by following Schüpbach's supplementary figures (Figs. Reply 1 and 2). Our results show rather greater contribution from northwestern Canada than that of Schüpbach's, but the entire distributions for both sites are similar each other and to that of Schüpbach's map. Thus, we consider that the method and results of our trajectory analysis can be interpreted as same as those of Schüpbach et al. (2018). In addition, Schüpbach et al. (2018) addressed a limitation of the back-trajectory analysis that could not capture the Asian dust transportation through the upper troposphere.

---

## Author Response (AR1)

Nagatsuka et al, "Variations in mineralogy of dust in an ice core obtained from northwestern Greenland over the past 100 years"

Response to reviewers' comments

Dear Editor, Dr. Elizabeth Thomas:

We appreciate all the reviewers and editor for taking the time to read our paper and many helpful suggestions. We consider the comments very carefully and will revise the manuscript. Below we provide point-to-point answers to the comments of the two reviewers and editor.

[Editor comment]

Thank you for your thorough response to the reviewer's comments. I am confident that you have addressed their concerns and your proposed changes will make substantial improvements to the paper.

Both reviewers identified the need for isotopic analysis to identify dust origin. I appreciate that this analysis requires substantial volumes and would not be within the scope of this paper. I believe you make note of this in the introduction but please ensure this is addressed in the revised manuscript. This would help explain why SEM-EDS is needed for low concentration Holocene sites.

[Author reply]

We have described that the Sr and Nd analyses need large amounts of samples, whereas the SEM-EDS can demonstrate a high-temporal-resolution record of composition and sources of ice core minerals during low dust concentration (Line 59-73).

[Editor comment]

Thank you for providing an in-depth explanation about the back-trajectories. I think the additional analysis will substantially improve the paper, providing context and explanation for the differences (and similarities) between SIGMA-D and the other sites. I would encourage you to include figure 5 (or a version of it) from your response in the updated version. For me, this really highlighted the different contributions by region. Please can you make the proposed changes and resubmit a revised version of the paper.

[Author reply]

Many thanks for this suggestion. We have added Figure 11 (shown as Fig. Reply 5) in the manuscript. We have also revised the words and/or sentences in the manuscript, which were highlighted in yellow.

Dear Dr. Anders Svensson as Reviewer #1

[Reviewer comment]
General comment: To investigate the dust source, it would however be very helpful to have a few isotopic analyses of Sr or Nd that are probably more indicative of the dust origin than the mineralogy and certainly more informative than eg the roundness parameter. The argumentation and thus the conclusions concerning the dust source being at high, mid or low latitudes based on mineralogy are in my view somewhat weak without the isotopes. Likewise, the lack of isotopic analyses makes comparison to similar analyses from many other Greenland ice cores impossible. Indeed, it would be a surprise if we have substantial amounts of low latitude dust deposited in Northern Greenland today.
[Author reply]
We have already replied to the reviewer comment and explained why we used not the Sr-Nd isotope, but the SEM-EDS analysis (Line 50-74). We have emphasized that the SEM-EDS analysis can demonstrate a high-temporal-resolution record of composition and possible sources of the ice core minerals during recent low dust concentration periods as the last century in the manuscript (Line 30-31 and 72-73).

[Reviewer comment] General comment: I have my doubts about the trajectory analysis suggesting that none of the dust in the ice core originates from Asia. Large Asian dust clouds are observed on satellite images following the dominating westerly wind pattern. During the last glacial the majority of the Greenland dust had this source. Likewise, several tephra studies show that volcanic material of high-latitude Asian and Alaskan origin makes it to Greenland, whereas no tephra from Europe or lower latitudes ever made it to Greenland to my knowledge. Today the major Greenland dust source is less well-known and a more local dust contribution certainly is a possibility, but I would suspect a fraction of the dust still to originate in Asia today. Without the isotopic tracers, however, it is difficult to know, and trajectory analysis may not tell the complete story. Please compare your trajectory analysis to that made in Schüpbach et al., 2018, and comment on differences in methodology and results. In particular, Schüpbach et al., 2018, supplementary figure 1 may be relevant.

[Author reply]
We have already replied to the reviewer comment about comparing our trajectory analysis with that made in Schüpbach et al., 2018. The sentences describing a limitation of the back-trajectory analysis has been added in Line 361-364.
We have also added sentences (Line 367-374) and figure (Fig. 11 also shown in this reply as Fig. Reply 5) to describe low contribution of air masses from eastern Asia and a similar

temporal change among the ice core sites, whereas the air mass contribution from the Greenland coast is high, especially for Hans Trausen and Renland, and followed by Sigma-D in the manuscript. The location map of the six ice core sites for the back-trajectory comparison has been added in Fig. 1.

[Figure]

Figure Reply 5. Air mass contribution for the 7-day back-trajectory from (a) Arid regions (China, Central Asia, and the Middle East), (b) North America (Canada and US), (c) Eurasia (EU and Russia), and (d) Greenland coast at the seven ice core sites on Greenland. Error bars indicate the standard deviation of contributions from 1981 to 2010. The analysis setting is same as Fig. Reply 4.

[Reviewer comment]

General comment: In Simonsen et al., 2019 (full reference below), the dust size distribution (or the contribution of large fraction particles) is seen as a strong indicator of local versus distant dust sources. Large particles cannot be transported over long distances, so the presence of large particles in the ice core is attributed to local dust sources. Does the dust size distributions in your study support your conclusions of local versus long-range dust source variations?

[Author reply]

Yes. Simonssen et al., 2019 used particles with diameters of <2 μm and >8 μm as indicators of the distant and local dust sources for the RECAP ice core, respectively. Size distribution of the SIGAM-D ice core dust showed that the samples from the cold period (1960 to 1989) contained less particles with diameters of >8 μm (0-2 particles) than those from the warm periods (1915 to 1959 and from 1990 to 2013, 1-9 particles) when local dust supply increased. We have added some sentences in the manuscript to describe above (Line 345-348).

We have also added the profiles of the SIGMA-D ice core dust concentration from 1915 to 2002 measured by an Abakus laser particle sensor connected to a continuous flow analysis (CFA) system (Line 350-356 and Fig. A3).

Co-authors who conducted the CFA analysis were added in Line 4-5 and Line 448.

[Reviewer comment]
General comment: I find the manuscript is generally well written with good figures and referencing, except the introduction that I think needs a rework both language-wise and concerning references. The cited papers appear not to be up to date; below I give some additional suggestions for recent work that I think should be referred to and possibly commented on.
[Author reply]
Many thanks for this suggestion. We have cited the papers as suggested by reviewers and get some English proofreading in the Introduction (Line 35-47, 54-63).

[Reviewer comment]
Mention the name of the analysed ice core in the title or in the abstract.
[Author reply]
We have added the name of the ice core in the abstract (Line 18).

[Reviewer comment] It would be relevant to cite this recent paper in the introduction, since the idea of more local dust in Greenland ice cores is also discussed here although on a different time scale:
Simonsen, M. F., Baccolo, G., Blunier, T., Borunda, A., Delmonte, B., Frei, R., Goldstein, S., Grinsted, A., Kjær, H. A., Sowers, T., Svensson, A., Vinther, B., Vladimirova, D., Winckler, G., Winstrup, M., and Vallelonga, P.: East Greenland ice core dust record reveals timing of Greenland ice sheet advance and retreat, Nature Communications, 10, 10.1038/s41467-019-12546-2, 2019.
[Author reply] Many thanks for this suggestion. We have cited this paper in the Introduction (Line 37, 63, 345-346).

[Reviewer comment] l. 19.: 'changed periodically' leaves the impression that the dust composition varies with a certain periodicity, ie with a certain frequency. Probably the authors mean to say that the dust had different composition in different periods?

[Author reply] We have revised the sentence that the ice core dust originated from different geological sources in different periods (Line 21-22).

[Reviewer comment] l. 23.: 'This indicates...' -> 'Comparison to Greenland surface temperature records indicates. . .'

[Author reply] We have revised the sentence as suggested by reviewer (Line 25).

[Reviewer comment] l. 24.: 'The trajectory . . .' -> 'A trajectory. . .' or 'Trajectory. . .'

[Author reply] We have revised the word as suggested by reviewer (Line26).

[Reviewer comment]

l. 30.: 'Past dust composition. . .' -> 'Past ice-core dust composition records have revealed substantial variations in the concentration...' It is unclear which time period you refer to here. Is this about glacial cycles (Antarctica), the last glacial cycle, the Holocene, historical times, or on a seasonal time scale? With this postulate you need to cite a reference.

[Author reply]

We have cited some papers and added sentences to refer to ice core dust variations on a geologic time scale in Greenland (Line 35-37).

[Reviewer comment]

l. 31-32.: 'climate signals'. Are you referring to ice core water isotopes? 'atmospheric circulation', what is referred to here? Again the time period is not specified. Again you will need to cite published work for this statement.

[Author reply]

We have revised the words 'climate signals, such as temperature and atmospheric circulation' to 'temperature changes' and add sentences to refer ice core dust records covering geologic time period (Line 38-47).

We cited some papers to refer correlated variations between Greenland ice core dust and temperature variability indicated by $\delta^{18}O$ records (Line 40-44).

[Reviewer comment]

l. 29-40.: I think you also need to cite at least some of those classic studies:

Ruth, U., Wagenbach, D., Steffensen, J. P., and Bigler, M.: Continuous record of microparticle concentration and size distribution in the central Greenland NGRIP ice core

during the last glacial period, Journal of Geophysical Research, 108, 10.1029/2002jd002376, 2003.

Petit, J. R., Mounier, L., Jouzel, J., Korotkevich, Y. S., Kotlyakov, V. I., and Lorius, C.: Palaeoclimatological and chronological implications of the Vostok core dust record, Nature, 343, 56-58, 1990.

Lambert, F., Delmonte, B., Petit, J. R., Bigler, M., Kaufmann, P. R., Hutterli, M. A., Stocker, T. F., Ruth, U., Steffensen, J. P., and Maggi, V.: Dust-climate couplings over the past 800,000 years from the EPICA Dome C ice core, Nature, 452, 616-619, 2008.

[Author reply]

Many thanks for this suggestion. We have cited these papers in the Introduction (Line 35-47).

[Reviewer comment]

l. 42.: What is meant by 'rarely'? Do you have evidence of changing isotopic rations during transport? If so, please cite relevant reference.

[Author reply]

We have used 'hardly alter' instead of 'rarely' in the sentence and cite some references (Line 51-52).

[Reviewer comment]

l. 41-50.: You probably should cite some of the more recent papers as well:

Újvári, G., Stevens, T., Svensson, A., Klötzli, U. S., Manning, C., Németh, T., Kovács, J., Sweeney, M. R., Gocke, M., Wiesenberg, G. L. B., Markovic, S. B., and Zech, M.: Two possible source regions for central Greenland last glacial dust, Geophysical Research Letters, 42, 10399-10408, 10.1002/2015GL066153, 2015.

Han, C., S. D. Hur, Y. Han, K. Lee, S. Hong, T. Erhardt, H. Fischer, A. M. Svens- son, J. P. Steffensen, P. Vallelonga. High-resolution isotopic evidence for a potential Saharan provenance of Greenland glacial dust. Scientific Reports 8:15582 | DOI:10.1038/s41598-018-33859-0 3, 2018. l. 51-60.:

Probably you should cite this paper:

Obbard, R. W., Baker, I., and Prior, D. J. (2011). Instruments and Methods A scanning electron microscope technique for identifying the mineralogy of dust in ice cores. Journal of Glaciology 57, 511–514.

[Author reply]

Many thanks for this suggestion. We have cited these papers in the Introduction (Line 55-59).

[Reviewer comment]

l. 63.: '... in the ice sheet' -> '... to the ice sheet'.

[Author reply]
We have revised the sentence as suggested by reviewer (Line 78).

[Reviewer comment]
l. 61-72.: This recent paper would also be relevant for the introduction:
Zhang, P., Jeong, J.-H., Yoon, J.-H., Kim, H., Wang, S.-Y. S., Linderholm, H. W., Fang, K., Wu, X., and Chen, D.: Abrupt shift to hotter and drier climate over inner East Asia beyond the tipping point, Science, 370, 1095-1099, 10.1126/science.abb3368.
[Author reply]
Many thanks for this suggestion. We have cited these papers in the Introduction (Line 80-82).

[Reviewer comment]
l. 179.: 'Dating of the SIGMA-D ice core, which was performed by annual layer counting of δ18O and Na+, showing obvious seasonal variations (Fig. A1).' -> 'Dating of the SIGMA-D ice core was performed by annual layer counting of δ18O and Na+ that show obvious seasonal variations (Fig. A1).'
[Author reply]
We have revised the sentence as suggested by reviewer (Line 211).

[Reviewer comment]
 l. 246.: 'cyclically' suggests a periodicity or frequency. What is the period or frequency on two different time scales referred to?
[Author reply]
We have revised the sentence that the ice core dust originated different geological sources in different periods (Line 281-282).

[Reviewer comment]
l. 250.: Are you suggesting a major contribution of low to mid-latitude dust to Northern Greenland for the period 1950-2004? What is the potential source area and why should it be turned on in 1950 and off in 2004?
[Author reply]
The SEM-EDS results imply that the possible sources of the ice core dust in 1950-2004 are likely to be low/mid latitudes because Type A mineral (kaolinite) that is typical of humid tropical climatic zones such as modern-day Africa, south America and southeast Asia (Line 148 and 169) is abundant in the period. Although trajectory analysis cannot estimate contribution from distant sources as described above, it is unlikely that the large amounts of ice core dust was transported from such tropical regions. On the other hand,

our trajectory analysis suggests that northern Canada can be also possible sources of the Type A mineral and some studies supported this argument. Thus, we concluded that northern Canada seems to be best candidate for the ice core dust source during the period (Line 375-387).

We have revised the manuscript to describe that the ice core dust was constantly supplied from a distant source (mainly northern Canada) to the SIGMA-D site, and was additionally provided from local ice-free areas in the warm periods (Line 388-393).

[Reviewer comment]
l. 309.: 'Previous studies. . .' What are those previous studies and are they concerned with dust transport to Greenland in the last century?
[Author reply]
We have cited references showing dust transported from Asian deserts during the last century from Greenland snow pit samples (Line 357-359).

Dear Dr. Laluraj C. M as Reviewer #2

[Reviewer comment]
General comment: The reporting of new dust mineral record observations in Greenland as well as attempts to identify the sources of dust during warmer/colder periods during 1915-2013 is worthy of publication. However, the report is not suitable for publication as it is. The strength of this work resides only in reporting a mineralogy record of a century time scale from Greenland; there have been no attempts to offer climatic reasons for recent changes in dust mineralogy/sources/production in source/deposition regions. Conversely, perhaps in the attempt to cover so much, the paper fails to go in-depth in a number of aspects of these important subjects and offer no insightful reasons for the possible causes in variation in dust mineralogy and sources changes during the recent colder/warmer periods of last century.

[Author reply]
We have already replied to the reviewer comment about climatic reasons for the temporal variations in mineral composition and sources of the ice core dust.

We have revised the manuscript to describe that the ice core dust was constantly supplied from a distant source (mainly northern Canada) to the SIGMA-D site, and was additionally provided from local ice-free areas in the warm periods (Line 388-393).

[Reviewer comment]
General comment:
In particular, the authors omitted the potential methods (stable isotope ratio of Sr and Nd) to identify the sources of dust in different time scales as well as failing to consider high-resolution fractional dust estimates using water-insoluble particle counting (Coulter counter and laser-sensing particle detector) along with the estimated mineralogy data for dust sourcing/variation of dust mineralogy for strengthening the present argument.

[Author reply]
We have already replied to the reviewer comment and explained why we used not the Sr-Nd isotope, but the SEM-EDS analysis (Line 50-74). We have emphasized that the SEM-EDS analysis can demonstrate a high-temporal-resolution record of composition and possible sources of the ice core minerals during recent low dust concentration periods as the last century in the manuscript (Line 30-31 and 72-73).

We have added the profiles of dust concentration from 1915 to 2002 measured by an Abakus laser particle sensor connected to a CFA system (Fig. A3), which supports an increase of local dust contribution from 1915 to 1950s as shown by the SEM results (Line 352-357).

[Reviewer comment]

General comment:

Further, lack of trajectories observed from well-known dust sources reported in Greenland (eg:-North Africa/Middle East, and Eastern Asia eg;- Simonsen et al., 2019 and publication by Svensson & group), thereby doubting the uncertainties in using reanalysis data prior to 1979 CE.

[Author reply]

See the reply to Reviewer #1.

[Reviewer comment]

General comment: Additionally, I strongly encourage the authors to consider the full ice core mineralogy/dust records of 223 m or/at least 113 m (1660-2013 CE) for a better understating of variation in mineralogy records in Greenland during a longer period instead of considering the present firn part of the ice core (38 m) where most of the uncertainties are present. I believe that the above-mentioned issues can be addressed with reasonable additions and extra analysis. I leave it to the Editor to decide whether to recommend, to modify/drop one of the subject study (mineralogy variation in Greenland during the last century) or leave them all.

[Author reply]

This paper aims to describe temporal variations in sources of minerals in a Greenland ice core focusing on the past 100 years when the Arctic region was remarkably warming. Furthermore, it takes a lot of time for the high-temporal-resolution SEM analysis. Thus, we have no plan to analyze the rest of the ice core dust samples in this study. We have added the sentences to describe above in the introduction (Line 86-87).

[Reviewer comment]

Line No. 23 - This indicates that the multi-decadal variation of the relative abundance of the minerals can be attributed to the local temperature changes in Greenland. The statement is quite arbitrary so may be re-written or removed.

[Author reply]

We have revised the sentences that the multi-decadal variation of the relative abundance of the minerals was likely affected by the local temperature changes in Greenland (Line 26).

[Reviewer comment]

Maybe a statement on the overall implication of your study or future perspectives will be useful at the end of the abstract.

[Author reply]

We have added the sentence "Our results suggest that SEM-EDS analysis can detect variations in ice core dust sources during recent periods of low dust concentration." at the end of the abstract (Line 30-31).

[Reviewer comment]
The first paragraph of the introduction may be rewritten or adding a new paragraph by including recent studies (eg:- Zhang et al., 2020; Bory et al., 2002 & 2003) on dust sources dust in Greenland/mineralogy of dust in Greenland/implications in terms of climate changes may be included.
[Author reply]
Many thanks for this suggestion. We have cited the references as well as those suggested by Reviewer #1 for ice core dust sources in Greenland (e.g. Újvári et al., 2015; Han et al., 2018; Simonsen et al., 2019)/ mineralogy and size of dust in Greenland (e.g. Ruth et al., 2003)/ implications in terms of climate changes (e.g. Petit et al., 1990; Lambert et al., 2008) and rewrited some sentences in the introduction.

[Reviewer comment]
Line No. 103- How reliable the SEM analysis is to estimate the particle size and the number of particle counting from ice core used in the present study? May be compared/clarified here using the different particle count estimate from other instruments
[Author reply]
We have already replied to the reviewer comment about size measurement and number counting of the ice core dust and described it in Line 127-129.
We have added the sentences to describe about the method and results of the dust concentration analysis (section 2.3 and Line 350-356).

[Reviewer comment]
Line No. 180- Since there are no other non-sea salt sulphate spikes of volcanic origin in the present time scale (1915-2013), Fig. A1 may be included with tritium spike data of chronology as a reference point
[Author reply]
We have added a tritium profile in Fig. A1.

[Reviewer comment]
Line 245- Is the abundance of silicate minerals due to volcanic ash? There are several volcanic dusts reported in Greenland ice cores during the period between 1915-2013. Please include a discussion on it.
[Author reply]

We have already replied to the reviewer comment.

[Reviewer comment]
Line No.310:-The choice of 07 days is not explained either and why the author considered only wet deposition?. At least this should be discussed.
[Author reply]
We have already replied to the reviewer comment about choice of 07 days.
We have added the sentences and figure describing mass contribution by dry deposition as well as wet deposition (Line 178-180, Figure 8).

[Reviewer comment]
Trajectories also can never be discussed without cross-checking the results with the general synoptic situation(s) during the transport.
[Author reply]
We have already replied to the reviewer comment.

[Reviewer comment]
Trajectories with links, as shown in Fig. 2, are very unlikely to represent the real path of the air/dust particle.
[Author reply]
We have already replied to the reviewer comment.

[Reviewer comment]
What does it mean by a little contribution from Asia? please quantify it.
[Author reply]
[Author reply]
We have already replied to the reviewer comment.

---

## Author Response (AR2)

Nagatsuka et al, "Variations in mineralogy of dust in an ice core obtained from northwestern Greenland over the past 100 years"

Response to editor's comments

Dear Editor, Dr. Elizabeth Thomas:
We appreciate you for taking the time out of your busy schedule to read our paper. Below we provide an answer to your comment.

[Editor comment]
Wording on line 365 – consider "….dust to an ice core sight is dependent on distance from the ice sheet margin…."

[Author comment]
We have revised the manuscript as suggested by editor.